# Structural variation, selection, and diversification of the *NPIP* gene family from the human pangenome

## Graphical abstract

## Authors

Philip C. Dishuck, Katherine M. Munson, Alexandra P. Lewis, ..., PingHsun Hsieh, Tomi Pastinen, Evan E. Eichler

## Correspondence

ee3@uw.edu

## In brief

*NPIP* is a highly expanded gene family in humans and African apes with extreme signatures of positive selection, yet its function and variation have not been characterized. Dishuck et al. use long-read assemblies and cDNA sequencing to identify structural polymorphism, ongoing positive selection, paralog-specific gene models, and tissue-specific expression.

## Highlights

- Positive selection observed in human nuclear pore interacting protein (*NPIP*) family

- *NPIP* duplications drive large-scale inversion polymorphisms and gene conversion

- Long-read transcript sequencing reveals tissue-specific expression of paralogs

- 56% of NPIP protein models have not been previously reported

Dishuck et al., 2025, Cell Genomics 5, 100977
October 8, 2025 © 2025 The Authors. Published by Elsevier Inc.

# Cell Genomics

CellPress

## Article

# Structural variation, selection, and diversification of the *NPIP* gene family from the human pangenome

Philip C. Dishuck,[1] Katherine M. Munson,[1] Alexandra P. Lewis,[1] Max L. Dougherty,[1,7] Jason G. Underwood,[1,2] William T. Harvey,[1] PingHsun Hsieh,[1,3] Tomi Pastinen,[4,5] and Evan E. Eichler[1,6,8,*]

[1]Department of Genome Sciences, University of Washington School of Medicine, Seattle, WA 98195, USA
[2]Pacific Biosciences (PacBio) of California, Incorporated, Menlo Park, CA 94025, USA
[3]Department of Genetics, Cell Biology, and Development, Institute for Health Informatics, University of Minnesota, Minneapolis, MN 55108, USA
[4]Genomic Medicine Center, Department of Pediatrics, Children's Mercy, Kansas City, KS 64108, USA
[5]UMKC School of Medicine, University of Missouri, Kansas City, Kansas City, KS 64108, USA
[6]Howard Hughes Medical Institute, University of Washington, Seattle, WA 98195, USA
[7]Present address: Tisch Cancer Institute, Division of Hematology and Medical Oncology, the Icahn School of Medicine at Mount Sinai, New York, NY 10029, USA
[8]Lead contact
*Correspondence: ee3@uw.edu

## SUMMARY

The *NPIP* gene family is among the most positively selected gene families in humans/apes and drives independent duplication in primate lineages. These duplications promote genetic instability, leading to recurrent disease-associated microduplication and microdeletion syndromes. Despite its importance, little is known about its function or variation in humans, as short-read sequencing cannot distinguish high-identity duplications. Using long-read assemblies of 169 human haplotypes, we find extreme variation in the content and organization of *NPIP* loci. We identify fixed and polymorphic paralogs and observe ongoing positive selection. With long-read RNA sequencing (RNA-seq), we create paralog-specific gene models, the majority of which were not previously documented, and observe paralog-specific tissue specificity. This analysis of an exceptionally dynamic gene family provides candidates for future functional study.

## INTRODUCTION

*NPIP* (also known as *morpheus*) is a gene family of unknown function that has undergone independent duplication in several primate lineages.[1–3] The gene family was first described based on the observation of a rapid expansion in African apes, where the underlying genes show a significant excess of amino acid replacements (extraordinary ratio of non-synonymous to synonymous substitutions [dN/dS] values) consistent with the action of positive selection.[1] The ~20 kbp duplicon that contains *NPIP*, LCR16a, is interspersed across human chromosome 16 (Figure 1A),[4,5] with a solitary copy on human chromosome 18.[1] These LCR16 duplications mediate recurrent duplications and deletions frequently associated with neurodevelopmental delay,[6–8] including one of the most common genetic causes of autism.[9,10] Altogether, the segmental duplications (SDs) associated with LCR16a[2] span ~10% of the euchromatic portion of human chromosome 16p, having emerged and expanded since ape divergence from the Old World monkeys (25 million years ago [mya]). The LCR16a-encoding *NPIP* has been described as a "core duplicon" for its characteristic overabundance within these intrachromosomal duplications.[11] It has independently

duplicated at least five times over the course of primate evolution, leading each time to the formation of interspersed SDs where lineage-specific duplications accrue flanking the core LCR16a duplicon.[2,3] Although the gene model has significantly changed among primates, the open reading frame (ORF) has been maintained over the course of primate evolution.[3]

Because *NPIP* is frequently embedded in large blocks of SDs that share >97% sequence identity (Figure 1), standard sequencing and assembly methods have limited our understanding of its genetic diversity and, consequently, our ability to make genetic associations or perform standard population genetic analyses. However, fluorescence *in situ* hybridization (FISH) analysis with LCR16a probes, along with read-depth analyses using short reads, have been used to estimate a range of 20–30 copies per human haplotype.[2,3] Targeted bacterial artificial chromosome (BAC) assemblies have partially resolved the *NPIP* loci in the most commonly used reference genomes. Because of the high sequence identity of the underlying duplicated segments, misassembly of the loci has been frequently encountered over the last 20 years of human genome assemblies. Even in one of the most recent human genome references, GRCh38, there is evidence of at least two chimeric

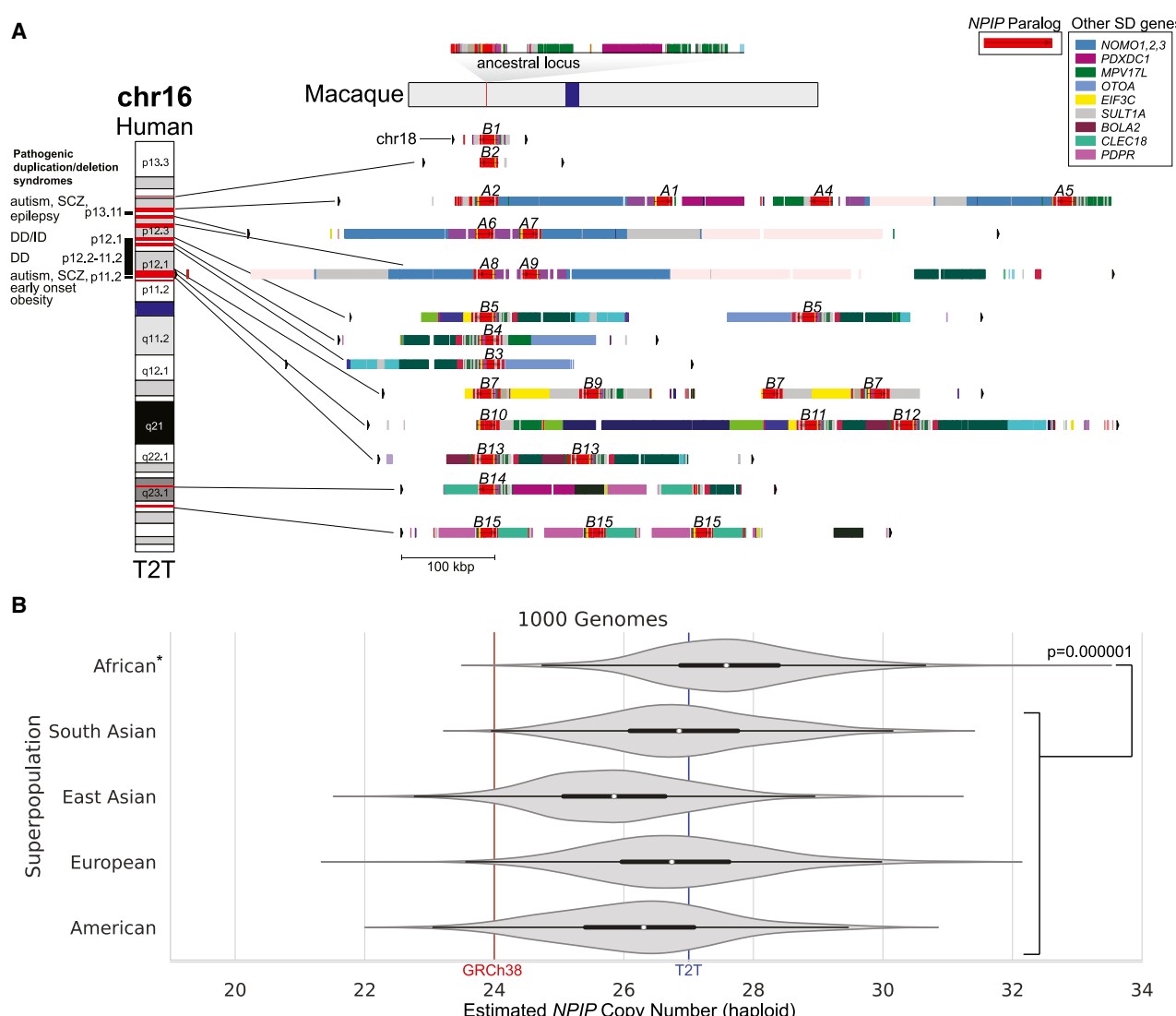

**Figure 1. NPIP locus organization and copy-number variation**

(A) NPIP regional organization in the T2T-CHM13 genome. The single-copy sequence in the macaque genome (Mmul10; top) is compared to the duplicated sequence on human chromosomes 16 and 18. Red highlights on chromosome 16 (left) correspond to the LCR16a duplicon encoding different NPIP genes, with segmental duplication (SD) content annotated by DupMasker (colored bars).[12] NPIP gene names (A1-9 and B1-15) are labeled above DupMasker tracks, with the colored bars indicating other chromosome 16 genes (key) associated with the NPIP expansion in humans. To the left of the ideogram, pathogenic duplication/deletion syndromes associated with NPIP SDs are shown as red horizontal bars on the ideogram (SCZ, schizophrenia; DD, developmental delay; ID, intellectual disability). These recurrent deletions and duplications are mediated by NPIP-containing SD blocks.

(B) Read-depth estimates (fastCN) of modern human per-haplotype NPIP copy number from the 1000 Genomes Project (n = 2,609), grouped by superpopulation. Copy number is significantly higher in African compared to non-African samples (Wilcoxon rank-sum test, p = 0.000001).

misassemblies being created due to inadvertent assembly of paralogous loci whose sequence identity approximates allelic variation. Not surprisingly, GRCh38 contains just 24 copies of LCR16a, compared to the median 25 copies estimated by short-read whole-genome sequencing (WGS) read depth to be present in most human haplotypes.[13]

Over the last few years, a series of resources and methods have been developed that make it possible to systematically characterize human genetic variation and expression across these regions of chromosome 16, arguably for the first time.

First, the T2T (Telomere-to-Telomere) Consortium recently completed the assembly of a single human haplotype, CHM13, by combining highly accurate, long HiFi (high-fidelity) reads with ultra-long ONT reads.[14] As a result, all NPIP gene copies are fully resolved in this haplotype, providing a complete reference for comparisons. Second, both the HPRC (Human Pangenome Reference Consortium) and HGSVC (Human Genome Structural Variation Consortium), using similar approaches, have published and released contiguous phased assemblies of 80 unrelated individuals.[15–17] The availability of both short-read

sequencing and long-read sequencing, including phasing information, enables the characterization and validation of entire chromosomal haplotypes for even the most identical gene families.[16,18,19] Third, the recent advancements of multiple sequence alignment (MSA) and phylogenetic methods optimized for comparing thousands of viral genomes[20,21] have facilitated the evolutionary reconstruction of rapidly evolving 20 kbp segments of human DNA, like LCR16a. We directly apply these methods to characterize thousands of *NPIP* paralogs and alleles to reconstruct the complex population genetic history underlying these regions of human chromosome 16, including the mutational forces that have shaped them. Finally, the recent release of 1.4 billion full-length cDNA from 384 isoform sequencing (Iso-seq) libraries from the Genomic Answer for Kids Study and ENCODE, among others,[22–36] makes it possible to assign transcript data to specific paralogs and alleles—a near impossibility previously with traditional short-read RNA sequencing (RNA-seq) data. We use these data to accurately construct gene models, define transcription start sites, distinguish potential protein-coding genes from pseudogenes, and interrogate expression and population genetic properties for specific *NPIP* copies.

## RESULTS

### Human genetic diversity

Using the complete sequence of the T2T human genome assembly, we first annotated LCR16a and its associated SDs using DupMasker for the T2T-CHM13 reference genome (Figure 1A). The analysis reveals 27 *NPIP* genes—26 of which map to 12 duplication blocks on chromosome 16 and a solitary copy mapping to chromosome 18, as expected.[1] This is in stark contrast to the macaque genome (Figure 1A), where only a single copy of LCR16a was identified. We assigned gene names based on the best matches according to the GRCh38 gene annotation. To estimate the copy-number distribution in the human population, we mapped whole-genome shotgun sequencing data[37] from the 2,609 unrelated individuals from the 1000 Genomes Project (1KG) using read depth to estimate the diploid and haploid copy number across each superpopulation. Among humans, we estimate that the haploid copy number ranges from 21 to 33 copies, with the highest copy number observed among individuals of African descent (Wilcoxon rank-sum test, $p$ = 0.000001; Figure 1B).

To understand the variation in structure of *NPIP* loci across the human population, we collected previously assembled and released genomes from the HPRC ($n$ = 43) and HGSVC ($n$ = 37),[15–17] along with a draft T2T assembly of HG002,[38] two individuals from Papua New Guinea,[39,40] the reference genome T2T-CHM13 v.2.0,[14] GRCh38, and an additional haploid cell line CHM1,[41,42] for a total of 169 unrelated haplotypes (Table S2). We identified and extracted *NPIP* loci from each assembly by aligning the *NPIP* locus from GRCh38 to each haplotype with minimap2 and wfmash (STAR Methods), for a total of 4,665 copies of *NPIP*. As *NPIP* loci are known to be structurally variable and subject to gene conversion, we did not rely on synteny alone to determine the paralog identity. Instead, we created an MSA and maximum likelihood phylogeny of the 4,665 *NPIP* loci from the 169 assembled haplotypes, using Bornean orang-

utan (*Pongo pygmaeus*) and siamang (*Symphalangus syndactylus*) as outgroups (Figure 2A). Monophyletic clades with >75% branch support (Shimodaira-Hasegawa-like approximate likelihood ratio test [SH-aLRT]) were used to assign copies to one of 28 defined paralogs, named based on phylogenetic identity to T2T-CHM13 and GRCh38. In cases where a clade did not have an anchor in T2T-CHM13 or GRCh38, we defined it based on its nearest neighbor (i.e., *NPIPA1L*). In cases where there was insufficient genetic distance to distinguish paralogs, they were grouped into a single clade comprising the two copies (i.e., *NPIPB12/B13*). For simplicity, we subsequently shorten gene names by dropping the *NPIP* prefix in this article. Additionally, to estimate the age of each branch, we created a time tree with LSD2 (STAR Methods),[43] incorporating paralogs from human assemblies CHM13, CHM1, GRCh38, HG002, and PNG15, along with nonhuman primate sequences from the T2T Primate Project (*Pan troglodytes*, *Pan paniscus*, *Gorilla gorilla*, *Pongo pygmaeus*, *Pongo abelii*, and *Symphalangus syndactylus*),[44] and the single-copy ancestral *NPIP* gene from *Macaca fascicularis*[45] as the outgroup (Figure S1).

Even among long-read sequence-assembled genomes, high-sequence-identity duplications that are hundreds of kbp in length remain a common source of misassembly and collapse.[46] We, therefore, validated the integrity of each assembled haplotype using computational tools designed to detect misassemblies (i.e., NucFreq, Flagger, and GAVISUNK[16,41,47]). For a haplotype structure to be classified as correctly assembled, we required contiguous assembly without collapse across all duplicated segments (not just *NPIP*) (Figure 1A), including at least 30 kbp of flanking unique sequences. The assembly validation rate varied from 52% to 85% (Figure 2A, right). The copy number of *NPIP* paralogs varies widely across assembled haplotypes (Figure 2B). Copy-number heterozygosity for the paralogs, defined as the frequency of discordant copy numbers between the two haplotypes of a sample, ranges from 0 for *B2*, *B11*, and *B14*, with all individuals having just one copy, to 0.74 for *A6/A9*. While only *B2*, *B11*, and *B14* are fixed in copy number, *A2*, *A4*, *B12/B13*, and *B15* always have at least one copy in all assemblies that pass quality control (QC). Individual members of *NPIP* subfamilies *B3-B5* and *B6-9* are not always present when considered individually, yet at least one paralog from each of these larger subfamilies is always present in a given haplotype (Figure 2B).

In addition to copy-number variation, interlocus gene conversion (IGC) is another common source of *NPIP* variation, as the high sequence identity among paralogs enables the replacement of the *NPIP* sequence from one paralog to another. As a result, the sequence content of a paralog does not always correspond to the same syntenic location when compared to other human haplotypes. We reanalyzed a recent genome-wide IGC callset for a subset of these haplotypes ($n$ = 94) to classify IGC patterns among *NPIP* duplication blocks.[48] As expected, IGC between paralogs is frequent (exceeding >50% of haplotypic configurations) and is driven primarily by proximity (1–2 Mbp), with six distinct IGC "hotspots" identified (Figures S2 and S3A) on the short arm of human chromosome 16. Active sites of gene conversion frequently correspond to the breakpoints associated with recurrent human microdeletion and microduplication

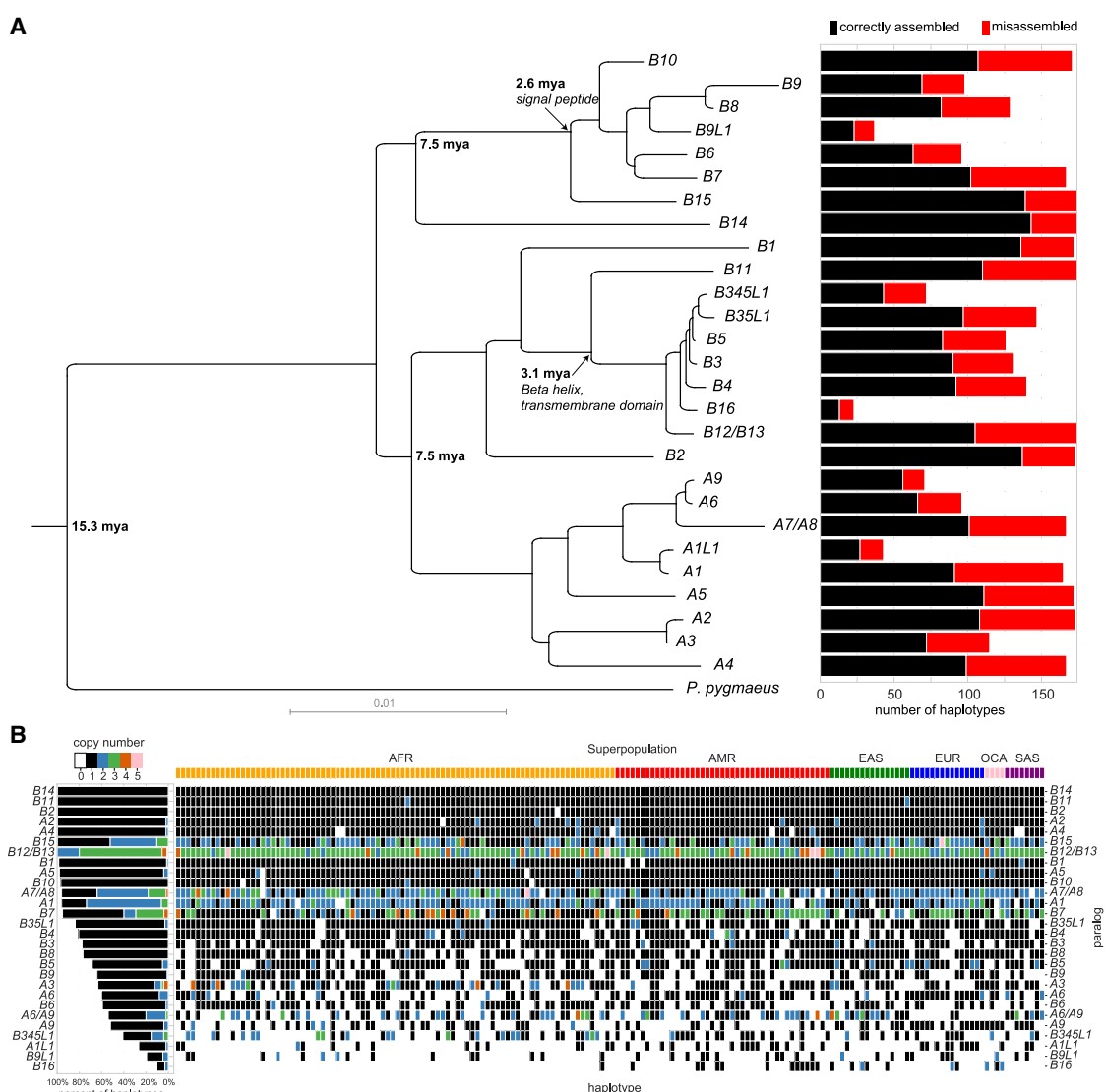

**Figure 2. Classification of human *NPIP* haplotypes and locus-specific copy number**

(A) Left: maximum likelihood phylogeny of human *NPIP* loci constructed using *Pongo pygmaeus* as an outgroup. It is based on 15 kbp of intronic (noncoding) sequence. Right: frequency of each paralog among the 169 haplotypes passing assembly validation (black) and number of misassembled loci where a potential collapse was typically identified (red).

(B) Copy-number summary of 169 assembled haplotypes. Color indicates the copy number of each gene, as defined by the phylogenetic grouping (A). Paralogs are sorted by fraction of validated haplotypes containing at least one copy. Left: percentage of haplotypes with each copy-number state for each paralog, restricted to assembled regions passing QC. Right: copy-number states for all assembled haplotypes, with each column representing a separate haplotype. Haplotypes are grouped by continental superpopulation (on top) (AFR, Africa; AMR, the Americas; EAS, East Asia; EUR, Europe; OCA, Oceania; SAS, South Asia).

syndromes (Figure 3A). IGC occurs within, but not between, the two major subfamilies (*NPIPB* copies undergo IGC only with *NPIPB* but not *NPIPA* loci). We also observe particular biases in donor/acceptor directionality. For example, the putative ancestral paralog *NPIPA1* acts only as a donor to *A5, A6, A8*, and *A9* locations in distal chromosome 16p but never as an acceptor, reflecting either functional constraint or bias in the mutation process itself.

During our comparative analysis, we frequently noted that the gene order of unique (nonduplicated) genes bracketed by *NPIP*

copies is inverted in different human haplotypes. Across chromosome 16p, we identify four inversion polymorphisms ranging in size from 350 kbp to 1.6 Mbp (Figures 3B–3E). The breakpoints of these inversions map either at *NPIP* copies or at associated SDs flanking *NPIP.* All of these large inversions are common polymorphisms (>5% allele frequency [AF]) and, in some cases, represent the major allelic configuration in the human population.[66,67] In several cases, the inverted unique sequence shows considerable allelic divergence (>0.2%), suggesting a deep coalescence, as has been observed for other human inversion

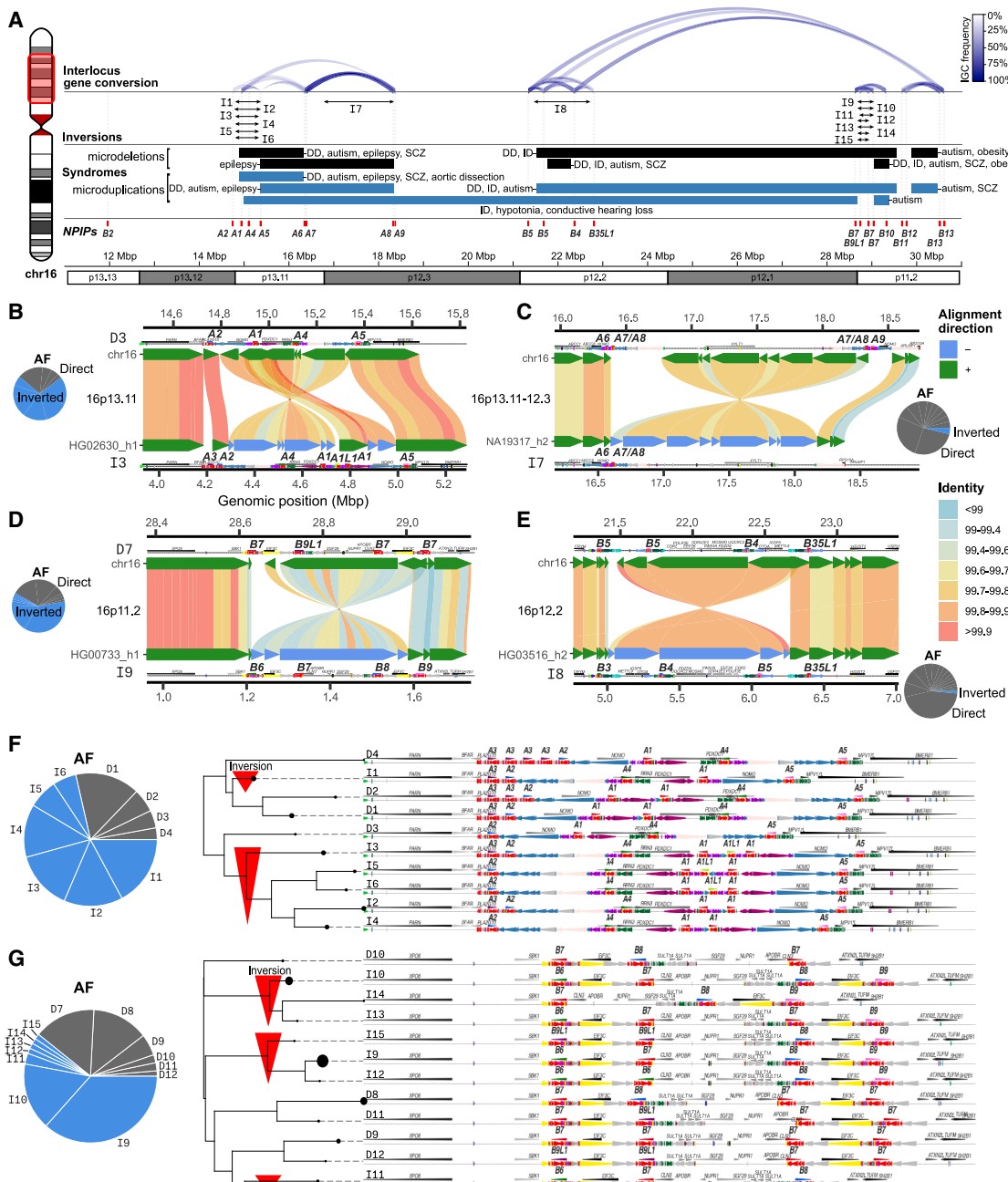

**Figure 3. *NPIP* interlocus gene conversion and complex structural changes**

(A) Overview of *NPIP* loci on chromosome 16p (highlighted ideogram region). The location of each T2T-CHM13 *NPIP* paralog is shown as a red vertical bar. The count and location of interlocus gene conversion (IGC) between *NPIP* pairs is shown as blue arcs at the top, with opacity corresponding to the number of observed haplotypes. Inversions mediated by *NPIP* are shown as black arrows, named corresponding to structures in (B)–(G). Known pathogenic microdeletions and microduplications with breakpoints at *NPIP* are shown as black and blue bars, respectively.[6–10,49–65]

(B–E) Large-scale inversion polymorphisms associated with *NPIP* loci (bottom) as compared to T2T-CHM13 v.2.0 (top). Inversions are shown with SVbyEye, with DupMasker annotations for each haplotype. Allele frequency (AF) for inverted (I) and direct (D) orientation haplotypes are shown with the pie charts.

(F and G) The duplication architecture (DupMasker) of the *A1-5* and *B6-9* loci for the most common haplotype configurations, grouped by a neighbor-joining tree of double-cut-and-join edit distance (pairwise number of rearrangements between configurations). The *NPIP* sequence is denoted in red. The size of the circle for each clade corresponds to the frequency of each configuration, and direct and inverted orientation configurations are named by frequency. Red arrows under the cladogram indicate configurations inverted with respect to T2T-CHM13.

polymorphisms on other chromosomes.[66,68] Indeed, the coalescence of the D7 inversion polymorphism at 16p11.2 was previously estimated as 1.35 mya and associated with susceptibility to asthma and obesity.[69,70] González et al. estimated that at least six distinct haplotypes exist at this locus based on multidimensional scaling of single-nucleotide polymorphisms (SNPs). Using phased genomes, we double this number, resolving 13 structural configurations at this locus, distinguished by orientation, *NPIP* paralog identity, and *SULT1A* copy number that were previously indistinguishable. This complete sequence resolution may help explain their observed association of the inversion with increased *SULT1A4* expression and decreased *SULT1A1* expression.[69] Notably, we find that many of the human haplotypic configurations occurred in conjunction with copy-number variation and IGC events associated with specific *NPIP* loci. A complete assessment of copy-number variation of both "unique" and SD genes flanking *NPIP* loci in disease regions may be found in the supplemental information (Table S4; Figure S7).

To more systematically classify different structural configurations, we encode haplotypes by the identity, order, and orientation of *NPIP* paralogs and marker genes. We apply a double-cut-and-join rearrangement distance metric (STAR Methods)[71] between each configuration to create corresponding neighbor-joining trees for each of the major *NPIP* clusters (Figures 3E and 3F). For example, at the ancestral chromosome 16p13.11 locus, we observe a 545 kbp inversion and the variable presence or absence of *A3* and a newly discovered paralog, *A1L1*. By contrast, *A5* maps invariably at the proximal end of this cluster (Figures 3A and 3E). The *A1L1* paralog only associates with 16p13.11 haplotypes that are inverted relative to T2T-CHM13; this 545 kbp inversion is the major allele (AF = 0.69). The chromosome 16p11.2 locus contains *NPIPB6*, *B7*, *B8*, and *B9*, spanning a 650 kbp inversion polymorphism (Figures 3D and 3G). Through IGC and inversions, the *B7* sequence can occupy any of the four canonical *NPIP* locations in this cluster—thus effectively "relocating" or "repositioning" as a result of IGC. Additionally, 8/13 configurations also have a 355 kbp inversion with respect to T2T-CHM13, and only the inverted orientation configurations carry the *B6* or *B9* gene. At 99.6% sequence identity to the reference genome, the I9 inverted region is among the most divergent (top 9.5%) euchromatic regions of the human genome. Of note, all direct orientation haplotypes (relative to T2T-CHM13) exhibit *NPIP* repositioning best explained by IGC, in contrast to 9% of inverted haplotypes, perhaps indicating that these paralogs underwent a period of diversification in the inverted orientation. We also observed 1.6 and 1.3 Mbp inversions at chromosomes 16p12.3 and p12.2, respectively (Figures 3D and 3E). Altogether, the results show that, of the nine loci containing *NPIP* paralogs, only the locus at chromosome 16p13.3 containing *B2* is structurally invariant.

### Diversity-based tests of selection
The *NPIP* gene family members were previously shown to harbor a significant excess of amino acid replacements, exhibiting one of the most extreme signals of positive selection in the human and African ape lineage (i.e., dN/dS > 1.0).[1] To assess whether positive selection is still ongoing in the human population and narrow down signatures to individual loci, we performed com-

plementary tests of Tajima's D and $nS_L$ for extended haplotype homozygosity,[72,73] restricting our analysis to chromosome 16. Tajima's D compares the number of segregating sites to pairwise differences to find deviations from the neutral expectation; negative values correspond to an abundance of rare alleles and are consistent with positive selection, while positive values correspond to a scarcity of rare alleles and are consistent with balancing selection. $nS_L$ (the number of segregating sites by length) is a test of extended haplotype homozygosity designed to detect recent hard and soft selective sweeps[73] and is more robust than Tajima's D to artifactual signals arising from bottlenecks and population growth. Unlike other haplotype-based tests of selective sweeps like integrated haplotype score (iHS), it is robust to phasing errors and does not rely on detailed recombination maps, as such maps are either nonexistent or unreliable in SD regions (STAR Methods).[74]

Previous attempts have been confounded by the inability to align short reads to these duplicated regions, but our contiguous haplotype-resolved assemblies now allow us to investigate whether there is evidence of selective sweeps across these regions in the human population. We calculated $nS_L$ and Tajima's D with HiFi assemblies, restricting to individuals of African ancestry, for whom we have the most samples of any individual superpopulation and to avoid bias due to the out-of-Africa bottleneck and subsequent expansions.[75] To evaluate the consistency of Tajima's D within unique sequence-flanking SDs, we compared Tajima's D using short reads from the 1KG (Gambian individuals). Signals are comparable in unique regions (Figure 4) but drop out over SDs for short-read sequence data. Using Tajima's D, we find negative values suggesting positive selection in the first percentile chromosome-wide for *NPIPB9*, *B12*, and *B15*, while *A1*, *A2*, *A5*, *B3*, *B4*, *B11*, *B13*, and *B14* are within the 5th percentile. *B7* and *A7* are, however, within the 5% most extreme windows for balancing selection (Figure 4A). Similarly, with $nS_L$, we find signatures of selective sweeps for *B7*, *B9*, and *B15* within the 1st percentile of most extreme values and for *A8* within the top 5th percentile (Figures 4C and 4D). The only two regions of consecutive $nS_L$ values in the first percentile correspond to *B7/9* and *B15 NPIP* loci. However, *B3*, *B4*, *B5*, *B7*, and *B9* are located within or near the boundaries of inversions, raising the possibility that suppressed recombination may be contributing to this signal (Figures 3D and 3E).

As Tajima's D has not previously been assayable within SDs, which exhibit an increased rate of IGC and mutations in general when compared to the unique regions of the genome,[48] we examined the empirical effect of IGC on Tajima's D values (Figures S6 and S8). We find that the Tajima's D distribution has an extended left tail when IGC-overlapping windows are included (Figure S6A), with the first percentile at −2.07 as compared to −1.87 with IGC-overlapping windows excluded. High-identity *NPIP* duplications experience high rates of IGC; indeed, the nearest IGC-free windows to each *NPIP* paralog (up to 289 kbp away) have less extreme Tajima's D values (Figure S6B). However, of the 11 *NPIP* paralogs within the 5% most extreme windows for positive selection, eight remain within the top 5% when IGC regions are excluded from this analysis: *A2*, *B9L1*, *B4*, *A7*, *A6*, *B14*, *B15*, and *B5*.

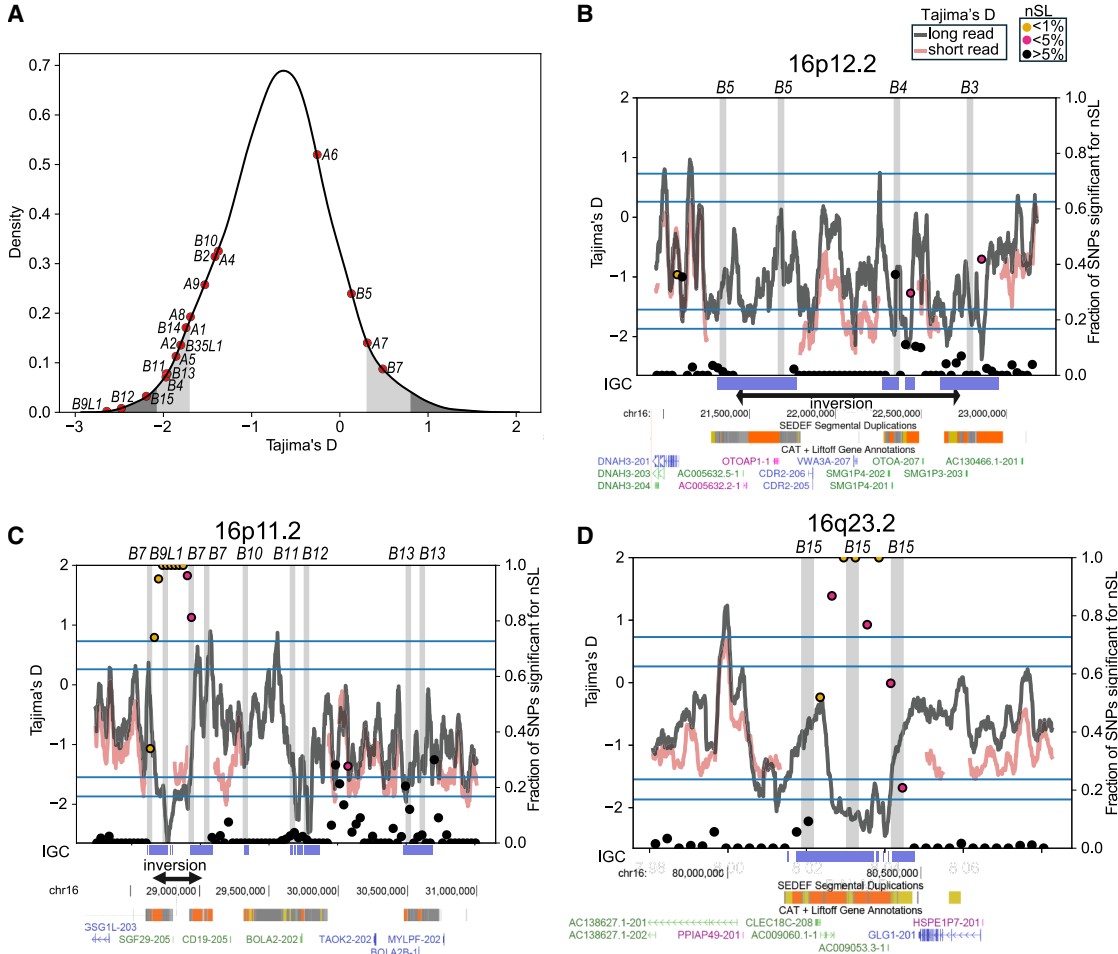

**Figure 4. Selection signatures at *NPIP* loci in the human population**

(A) Tajima's D distribution on chromosome 16, based on alignment of long-read sequence and assembled human haplotypes. The most extreme 1% and 5%, both positive (balancing selection) and negative (positive selection), are colored in gray and dark gray across the chromosome 16 distribution, with the values for specific *NPIP* paralogs windows highlighted (red dots).

(B–D) Results of Tajima's D and $nS_L$ selection tests for three loci showing signatures of positive selection. Short-read (red) and long-read (gray lines) Tajima's D results are shown. $nS_L$ values are plotted as filled circles, with color indicating significance. Known inversion polymorphisms are indicated (black arrows on the bottom) along with SDs and T2T gene annotations. Horizontal lines indicate 1% and 5% thresholds for Tajima's D, both positive and negative. Vertical gray highlights indicate locations of *NPIP* paralogs, with gene names and SDs (SEgmental Duplication Evaluation Framework [SEDEF]) shown on the bottom. Regions with detected IGC events are shown on the bottom in blue.

### *NPIP* gene models and differential tissue expression

Previous research demonstrated ubiquitous expression of *NPIP* paralogs in apes, as compared to the largely testis-specific expression in Old and New World monkeys, along with slightly different gene models for human *NPIPA* and *NPIPB* subfamilies.[3,76] With our more complete catalog of human *NPIP* paralogs, we sought to determine whether we could identify additional paralog-specific gene models and if there is evidence of tissue-specific expression when considering particular *NPIP* paralogs instead of the family as a whole. Short-read RNA-seq does not align uniquely to *NPIP* paralogs due to their high sequence identity, preventing the construction of complete gene models and paralog-specific expression estimates. Instead, we used PacBio HiFi sequencing of full-length cDNA

(Iso-seq), facilitating the unambiguous assignment of the majority of Iso-seq reads to specific *NPIP* paralogs. To this end, we assembled a database of full-length non-chimeric (FLNC) cDNA generated from 1.4 billion Iso-seq reads from 384 libraries, representing 101 human tissue and cell types (Table S1). To complement this effort, we also performed hybridization capture experiments against select tissues using *NPIP*-targeting capture probes in order to enrich for *NPIP* FLNC molecules (STAR Methods).[77] We extracted Iso-seq reads aligning to any *NPIP* paralog, totaling 1.07 million reads with an average length of 1,960 nt. To create paralog-specific gene models, we considered ORFs seen in at least five Iso-seq molecules as valid and only display the most abundant and longest isoforms for each paralog (Figure 5A).

Cell Genomics
Article

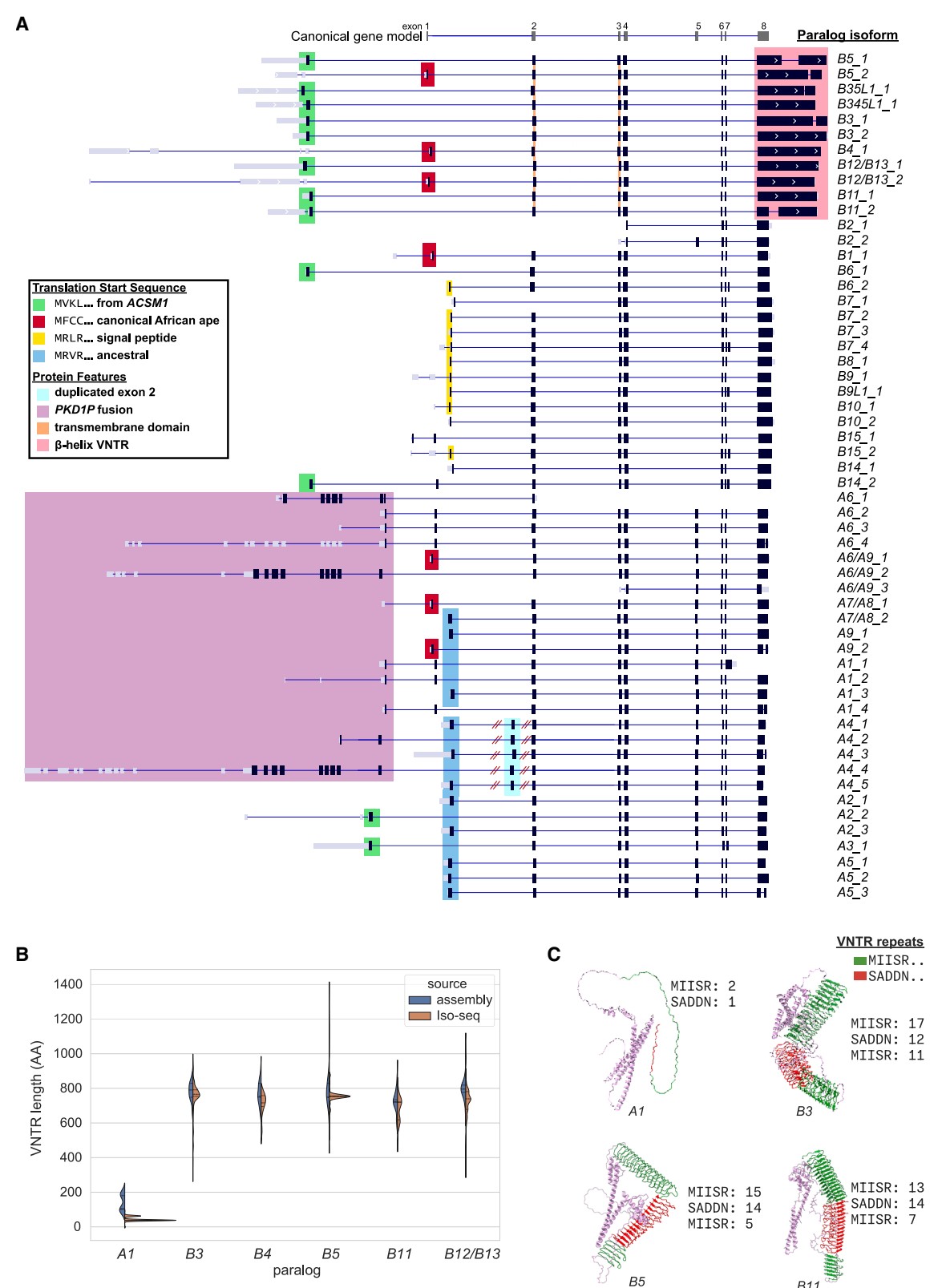

(legend on next page)

We observe Iso-seq molecules encoding full-length ORFs for most *NPIP* paralogs (Figure 5). This includes four paralogs that had previously been annotated as noncoding pseudogenes, *NPIPB1P*, *NPIPP1*, *NPIPB10P*, and *NPIPB14P*, which we refer to as *B1*, *A4*, *B10*, and *B14*, respectively. The African-ape-specific *B1* paralog, the only human paralog on chromosome 18 and therefore not predisposed to the same level of structural variation, was previously reported to neither be transcribed nor maintain an ORF.[3] By contrast, we find that it maintains an ORF and is expressed, albeit at low levels, in testis as well as brain organoids.

Closer inspection of these gene models reveals a considerable amount of variation in predicted amino acid composition across *NPIP* paralogs and their isoforms due to alternative promoters, differences in translation initiation, and expansion of protein-encoding variable number tandem repeats (VNTRs) at the C terminus. Consequently, ORFs range in length by 8-fold (155–1,217 amino acids [aa]). Of the 55 most common *NPIP* isoforms, only seven begin with the canonical first coding exon "MFCC …," which is shared with African apes,[3] and only 24 of these 55 were represented in RefSeq, allowing for amino acid substitutions and VNTR variation. Eleven begin with an alternate translation initiation "MVKL" sequence, previously identified as the start sequence for the *NPIPB* subfamily.[76] In addition to *NPIPB* paralogs, we also observe this start sequence for *NPIPA2* and *A3* and determine that this 40 aa exon arose from an independent duplication of the twelfth exon of *ACSM1*, an acyl-coenzyme A (CoA) synthetase gene (Figure S3), including half of its AMP-binding enzyme C-terminal domain (InterPro domain IPR025110). Cantsilieris et al. reported an "MRVR" start sequence in non-African ape primates, perhaps the ancestral sequence; we observe this start site used in 11 human *NPIPA* isoforms. A subset of *NPIPB* members (*B6*, *B7*, *B8*, *B9*, *B10*, *B14*, and *B15*) use a previously undocumented "MRLR" start site, encoding a 19–26 aa signal peptide, as predicted by SignalP-6.0.[78] Though the sequence that encodes the signal peptide is present in all human *NPIP* paralogs and shared with nonhuman primates, we estimate the clade that uses this sequence as its transcription start site to be human specific, arising ~2.6 mya during the evolution of our lineage.

Finally, for the six *NPIP* paralogs adjacent to *PKD1* pseudogenes (*A1*, *A4*, *A6*, *A7*, *A8*, and *A9*), we observe 10 distinct *PKD1-NPIP* fusions, four of which are multi-exonic, linking up to nine *PKD1* exons (530 aa) with eight *NPIP* exons (343 aa). Remarkably, these fusions maintain long *PKD1-NPIP* ORFs up to 843 aa in length. *PKD1* variants are implicated in polycystic kidney disease, as well as estimated glomerular filtration rate.[79] Though the *NPIP*-adjacent *PKD1* copies have been considered pseudogenes because the *PKD1* duplications are truncated and do not encode full-length genes,[80] there is precedent for truncated genes to be functional. Several partial gene duplications like *SRGAP2C*, *NOTCH2NL*, and *ARHGAP11B* have been shown to be functional through dominant-negative interactions.[81–83] The role of these *PKD1-NPIP* fusion transcripts is unknown.

The final coding exon of the human-specific *NPIPB* subfamily (*B3*, *B4*, *B5*, *B11*, *B12*, and *B13*) contains an expanded in-frame VNTR. Our analysis of 169 haplotypes and hundreds of cDNA libraries demonstrates that even within single paralogs, the copy number of this VNTR is variable among individuals. The VNTR encodes a repetitive amino acid motif of 19 (SADDNLKTP SERQLTPLPP) or 23 (SADDNIKTPAERLRGPLPPSAPP) residues, with the two lengths alternating. Within each paralog, the sequence frameshifts from the SADDN … form (7–15 repeats) to a MIISRHLPSVSSLPFHPQLHPQQMI form (6–14 repeats) and back to SADDN … (5–11 repeats) in the genomic annotations, resulting in a repeat domain ranging in size from 297 to 1,298 aa (Figure 5B). Analyzing Iso-seq cDNA directly, we observe 8,986 molecules sharing this VNTR switching pattern with up to 25, 20, and 17 repeat units. Computational protein structure prediction suggests that both frames of the VNTR may form a left-handed β helix, with each VNTR unit corresponding to an additional turn of the helix, kinked as the frame shifts between these two amino acid motifs (Figure 5C). The gene models for this subfamily also encode a transmembrane domain as predicted by DeepTMHMM.[84] We estimate that this specific gene and protein structure of *NPIP* is, once again, human specific, arising ~3.1 mya (Figures 2A and S1).

We attempted to assess paralog-specific expression levels of *NPIP* paralogs leveraging both Iso-seq reads and short-read RNA-seq using a unique k-mer approach to specifically tag the short-read data. To determine paralog identity, the 1.07 million *NPIP* Iso-seq reads were aligned to each of the 169 assembled haplotypes, recording the location of best mapping and requiring a difference of at least one additional mismatch to the next-best mapping paralog to consider a read uniquely identified. The approach allowed us to assign ~55.6% of Iso-seq reads to create paralog-specific gene models. Grouping highly similar paralogs (*A2/3*, *A6-9*, *B3-5*, and *B12/13*) allowed us to assign ~93.2% of Iso-seq reads for expression analysis. While these estimates are not quantitative due to errors and biases inherent in library preparation and sequencing, we observe relative and reproducible differences in paralog expression across tissue types. Comparing expression between tissue types, specific clusters of paralogs have increased relative expression in distinct tissues. In particular, *NPIPA1*, *A5*, *A6-9*, *B3-5*, and *B12/13* show increased expression in fetal or adult brain relative

**Figure 5. Paralog-specific gene models**

(A) Most common isoforms for each *NPIP* paralog based on full-length cDNA Iso-seq mapping. The _x suffix indicates relative abundance (i.e., B5_1 is the most abundant B5 gene model). Predicted protein-coding regions (black), untranslated regions (gray) with different protein start sequences, and structural features (color) are highlighted over the gene models, including transcripts with the expanded protein-encoding β helix (pink) and the signal peptide (yellow). The canonical *NPIP* gene model is depicted (top). See Table S5 for cDNA and predicted amino acid sequences of *NPIP* paralogs/isoforms.

(B) Comparison of VNTR length encoding the β helix of exon 8 in the genome assemblies versus Iso-seq data.

(C) Predicted protein structures for four paralogs with exon 8 VNTR sizes. The copy number of repeating amino acid motifs by type are indicated and projected onto Chai-1 structure predictions (MIISR … repeat protein domain shown in green, while frameshifted VNTR SADDN … repeat protein domain in red).

to other tissues, while *A2-3*, *A4*, *B1*, *B2*, *B6-9*, *B10*, *B14*, and *B15* retain the presumed ancestral testis-enriched expression pattern (Figures 6A and S4).[76] Immune function-related tissues like tonsil, B cells, granuloma, and blood also significantly over-express paralogs seen in brain or testis.

Finally, we also attempted to define developmental time-point specificity by classifying short-read RNA-seq reads from an atlas of organ development[85] based on the presence of uniquely identifying k-mers from the 169 haplotypes. Paralogs that contained few uniquely identifying k-mers were combined into larger paralog groups for this analysis (STAR Methods). Altogether, 25.2% of reads containing any *NPIP* k-mer (n = 1.06 million) were uniquely assigned to a paralog or paralog group. *NPIPA1*, *A4*, and *B3-5* tend to increase in expression in the cerebellum after birth (Figure 6B). In contrast, *B1*, *B2*, *B6-9*, *B10*, and *B15* expression is almost entirely testis specific, with levels increasing after puberty (Figures 6C and S5).

## DISCUSSION

The expansion of *NPIP* and its associated SDs across the short arm of chromosome 16 predisposes humans to frequent recurrent pathogenic duplications and deletions associated with autism, developmental delay, epilepsy, and obesity.[6–10,49–64] Despite this negative effect on fitness, the duplications not only persist but have expanded among the African great apes, albeit most often at non-orthologous locations.[2,3] Moreover, these same sites have been homogenized via IGC, ensuring that high sequence identity is maintained and driving high rates of non-allelic homologous recombination associated with disease. In light of the strong signals of positive selection for this hominid gene family,[1,3,39] we hypothesized that an evolutionary trade-off exists between disease susceptibility and as-of-yet unknown adaptive function(s). In this work, we catalog, using a pangenomic approach, normal human variation at each *NPIP* locus and identify paralog-specific features potentially relevant to understanding the function of this enigmatic and dynamic gene family.

Because functional human-specific duplicate genes have been shown to be frequently invariant,[81,82,86] we systematically assessed the copy number for each paralog. Based on our 28 distinct human phylogenetic groups, we find only three loci that are copy-number invariant (*NPIPB2*, *B11*, and *B14*). We also distinguish loci that always have at least one copy in humans, although they often have more (*A2*, *A4*, *B12/B13*, and *B15*). It should be noted that several of these copies are associated with larger-scale structural changes, such as inversion polymorphism or IGC events. Based on the human genomes we surveyed here, only the *NPIPB2* locus is invariant across all analyses, and counter to expectation, *NPIPB2* is the only paralog where all predicted ORFs are truncated, and it may be a *bona fide* pseudogene. Notably, this particular copy is the only paralog that is isolated (i.e., not found in a cluster with other *NPIP* paralogs nor associated with other flanking SDs). This stands in contrast to the ancestral locus, *NPIPA1*,[1–3] which is both copy-number polymorphic and shows a striking asymmetry for IGC, serving only as a donor and never an acceptor of a gene conversion event. We similarly observed asymmetric IGC in another recent human duplication, the *NOTCH2NL* gene family, which

is associated with the expansion of the human neocortex. In that case, 20% of haplotypes converted *NOTCH2NLB* to *NOTCH2NLA*, while the reciprocal event is never observed.[87] Such striking patterns of biased IGC may further pinpoint the copies most likely to confer function.

We previously identified extreme evolutionary signatures of positive selection for *NPIP* between African ape species, in particular, the *NPIPB* subfamily, based solely on tests for an excess of amino acid replacement among paralogs.[1,3] With highly contiguous haplotype-resolved assemblies, we were able to apply population-level selection tests for the first time. Using Tajima's D and $nS_L$, we find that *NPIPB9* and *B15* are within the top percentile of most extreme values for both tests on chromosome 16, while nine additional paralogs occur within the top 5th percentile for at least one test. We also find some evidence of balancing selection for a few loci (*A7* and *B7*). While these findings strongly suggest ongoing positive selection in humans, caution must be exercised given the large-scale structural changes and IGC associated with these regions. While specific copies drop out if we exclude regions of IGC, we continue to observe signals of positive selection (e.g., *NPIPB15*) in contemporary human populations, though we no longer find evidence of balancing selection (Figure S6). Additionally, *NPIPB3-B9* are located within or near the boundaries of inversions, raising the possibility that suppressed recombination may be contributing to this signal, including extended haplotypes (Figures 3D and 3E). These signals may not, however, be mutually exclusive with inversions enriched for adaptively evolving genes.[88,89] Such is the case of the 17q21.31 inversion polymorphism—a locus associated with increased fecundity,[68,90] positive selection in humans,[91,92] and the dynamic evolution of newly minted gene family *LRRC37A1/2*[93–95] expressed highly in human astrocytes.[96]

With a database of 1.4 billion FLNC reads (Iso-seq), we were able to comprehensively construct paralog-specific gene models, of which 56% (31/55 most abundant isoforms) had not been previously described in RefSeq. All but one paralog maintains a full-length ORF, while *B2*, the most copy-number invariant, is predicted to encode a truncated protein. The full-length gene models reveal new features—such as *NPIP* subfamilies gaining a start sequence co-opted from *ACSM1*, a novel signal peptide, transmembrane domain, or a variably sized coding VNTR that is predicted to form a β helix and, therefore, alter the protein structure. We estimate that the signal peptide and β helix evolved independently 2.6–3.1 mya and are innovations specific to the human lineage of evolution. The specificity afforded by long-read sequencing or paralog-specific k-mer analysis also reveals tissue-specific differences. For example, the subfamily encoding the novel signal peptide includes the two paralogs with the strongest signal of positive selection (*B6-9* and *B15*). This set shows testis-enriched expression, and analysis of a short-read development dataset additionally indicates that these paralogs increase in abundance at the onset of puberty. The paralogs with the novel β helix and transmembrane domain, by contrast, tend to be enriched in brain samples (*B3-5* and *B12/B13*), with *B12* also among the strongest signals of positive selection.

In summary, the dynamic changes in copy number, gene model, and expression specificity across *NPIP* paralogs, along with strong signals of positive selection, suggest

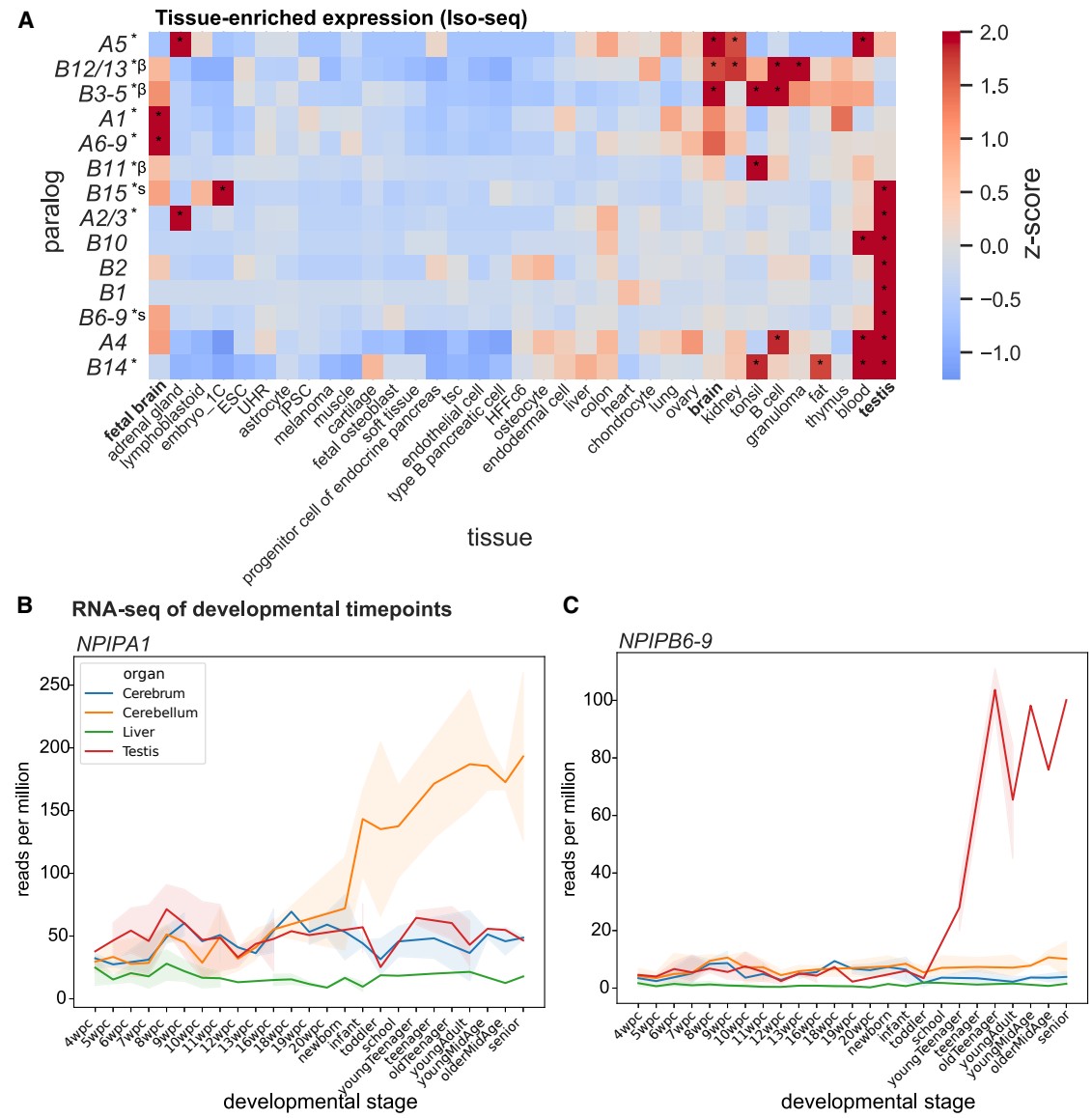

**Figure 6. Variable expression of *NPIP* paralogs across tissues, cell types, and developmental time points**

(A) Relative enrichment of Iso-seq expression estimates for 35 tissues, clustered with unweighted pair group method with arithmetic mean (UPGMA). Significantly positive Z scores are indicated, *$p < 0.05$. Paralogs with selection signatures are indicated with * at left. β, β helix; s, signal peptide.

(B and C) Short-read RNA-seq expression estimates for human developmental time points in four tissues for *NPIPA1* and *NPIPB6-9* paralogs (aggregate), using unique k-mers for paralog identity. Transparent error bands represent 95% confidence interval of replicates.

neofunctionalization of specific copies during human evolution. Both the paralogs that have maintained the presumed ancestral testis expression pattern (*B15*) and those that have gained enriched brain expression (*B12/B13*) exhibit clear signatures of positive selection. Concurrently, these two subfamilies evolved radically distinct gene models and associated protein structural changes in the human lineage. All of these changes have occurred and potentially been accelerated in a milieu of recurrent structural variation and IGC. Notwithstanding, it is noteworthy that *B15* and *B12/B13* are among just four paralogs that are sometimes duplicated but never deleted among humans, a po-

tential indication of their intolerance to loss. Now that the organization and variation of these oft-overlooked loci have been resolved and their variation and gene structures understood as part of human pangenomic efforts, the next step will be associating this variation with human phenotypes, including disease.

## Limitations of the study

Many of the new gene models require validation at the level of the protein, as proteomic data in support of these are sparse. Distinguishing protein paralogs that differ by a few amino acids has been challenging, but it is now potentially possible given that

the long-read transcriptomic data predict significant N and C termini differences. Protein features specific to a subset of paralogs, including the fusion/hybrid genes or the expanded VNTR predicted to encode a β helix in a subset of human-specific *NPIPB* members, may provide useful markers to test by mass spectroscopy for the presence of *NPIP* proteins in tissues. Another limitation of this work is the inability to assign all of the long-read transcriptomic sequence data to the most appropriate paralog. The perfect sequence identity of duplicates that approaches polymorphism levels and the rampant gene conversion, structural variation, and copy-number variation that exist among humans result in a considerable fraction of unambiguous assignments. A potential solution to this problem would be the generation of DNA-RNA matched resources where a T2T genome is fully resolved, as well as long-read transcriptomic and methylation data from the same individuals generated. The development of such resources as part of the Somatic Mosaicism Across Human Tissues project[97] provides an opportunity to begin to interrogate the tissue expression profiles of individual members of recently duplicated gene families more systematically. Finally, while we have detected signatures of positive selection among humans and between ape species, the nature of this selective force is unknown, in large part because the function of this gene family is still a mystery. There have been suggestions that the protein interacts with members of the nuclear pore complex[1] and that high expression in the macula of the retina may implicate some members in playing a role in visual acuity.[98] These suggestions are, however, speculative, requiring further functional characterization. Mice knockin experiments[3] have not revealed an overt phenotype. Discovery of a human phenotype associated with disruption of these genes would be the most informative, albeit challenging. Although many of the genomic disorders mediated by *NPIP*-associated rearrangements associate with autism and developmental delay,[65] the phenotypic consequences are more likely to result from the deletion or duplication of unique genes flanked by the SDs as opposed to dosage changes in *NPIP* copy number. Now that individual paralogs can be sequenced and assembled from controls, it should, however, be possible to identify individuals with specific mutated copies using long-read sequence data.

## RESOURCE AVAILABILITY

### Lead contact
Further information and requests for resources and reagents should be directed to and will be fulfilled by the lead contact, Evan E. Eichler (ee3@uw.edu).

### Materials availability
This study did not generate new unique reagents.

### Data and code availability
- Data are available at the accessions listed in Tables S1 and S2.
- This paper does not report original code.
- Any additional information required to reanalyze the data reported in this paper is available from the lead contact upon request.

## ACKNOWLEDGMENTS

We thank Tonia Brown for assistance in editing this manuscript. This work was supported, in part, by a US National Institutes of Health (NIH) grant (R01HG002385) to E.E.E. and an NIH Pathway to Independence Award (5R00HG011041) to P.H. E.E.E. is an investigator of the Howard Hughes Medical Institute (HHMI). This article is subject to HHMI's Open Access to Publications policy. HHMI lab heads have previously granted a nonexclusive CC BY 4.0 license to the public and a sublicensable license to HHMI in their research articles. Pursuant to those licenses, the author-accepted manuscript of this article can be made freely available under a CC BY 4.0 license immediately upon publication.

## AUTHOR CONTRIBUTIONS

Conceptualization, P.C.D. and E.E.E.; methodology, P.C.D., K.M.M., M.L.D., J.G.U., W.T.H., P.H., and E.E.E.; investigation, P.C.D., K.M.M., A.P.L., M.L.D., J.G.U., and W.T.H.; writing, P.C.D. and E.E.E.; funding acquisition, E.E.E. and P.H.; resources, T.P. and E.E.E.; supervision, E.E.E. and P.H.

## DECLARATION OF INTERESTS

E.E.E. is a scientific advisory board (SAB) member of Variant Bio, Inc.

## STAR★METHODS

Detailed methods are provided in the online version of this paper and include the following:

- KEY RESOURCES TABLE
- METHOD DETAILS
  - Short-read copy number estimation
  - *NPIP* gene identification
  - Other gene annotation
  - Phylogenetic paralog identity
  - Assembly validation
  - Locus configuration comparisons
  - VNTR analysis
  - Gene model and ORF prediction
  - Selection analyses
  - Protein structure prediction
  - Short-read RNA-seq expression analysis
  - Visualization
  - Timetree analysis
  - Probe design, cDNA generation, enrichment, and sequencing
- QUANTIFICATION AND STATISTICAL ANALYSIS

## SUPPLEMENTAL INFORMATION

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

**Cell Genomics**
Article

## STAR★METHODS

### KEY RESOURCES TABLE

| REAGENT or RESOURCE | SOURCE | IDENTIFIER |
|---|---|---|
| **Biological samples** | | |
| CHM1 | Magee-Womens Hospital | SAMN02205338 |
| Adult brain | Clontech | 636102 |
| Fetal brain | Clontech | 636106 |
| Heart | Takara | 636532, lot 1902102A |
| Lung | Origene | FR5B3386C1 |
| Ovary | Origene | FR00027E9B |
| Thymus | Origene | FR5B338054 |
| Testis | Takara | 636533, lot 1402004 |
| **Deposited data** | | |
| Whole transcriptome and targeted Iso-Seq data – see Table S1 | This paper and accessions listed in Table S1 | https://doi.org/10.5281/zenodo.14941577 |
| **Oligonucleotides** | | |
| *NPIP*-targeting hybridization probes used for cDNA enrichment – see Table S3 | This paper | N/A |
| **Software and algorithms** | | |
| fastCN | Pendleton et al., 2018[97] | https://github.com/KiddLab/fastCN |
| Wfmash v0.7 | Marco-Sola et al., 2023[65] | https://github.com/waveygang/wfmash |
| Minimap v2.22 | Li, 2018[12] | https://github.com/lh3/minimap2 |
| Liftoff v1.6.3 | Shumate and Salzberg, 2021[99] | https://github.com/agshumate/Liftoff |
| MAFFT v7.487 | Katoh and Standley, 2013[99] | https://mafft.cbrc.jp |
| IQ-TREE v2.2.3 COVID-edition | Minh et al., 2020[100] | https://github.com/iqtree/iqtree2 |
| SQANTI3 v5.2 | Parco-Palacios et al., 2024[101] | https://github.com/ConesaLab/SQANTI3 |
| Chai-1 | Chai-Discovery et al., 2024[102] | https://github.com/chaidiscovery/chai-lab |
| Hisat2 | Kim et al., 2019[103] | https://daehwankimlab.github.io/hisat2/ |
| Jellyfish | Marçais and Kingsford, 2011[100] | https://github.com/gmarcais/Jellyfish |
| LSD2 | To et al., 2016[43] | https://github.com/tothuhien/lsd2 |

## METHOD DETAILS

### Short-read copy number estimation

We applied fastCN to high-coverage Illumina data for 2,609 unrelated individuals from the 1KG to estimate *NPIP* copy number.[37,104,105] Short-read shotgun sequences from each individual are split into 36 bp segments and aligned to a reference genome (up to two single-nucleotide mismatches) allowing copy number to be estimated.[105] Windows overlapping the exon 8 VNTR were excluded from copy number estimation to avoid biasing the estimate.

### *NPIP* gene identification

To identify *NPIP* gene locations within assemblies, we aligned the ancestral *NPIP* locus from GRCh38 (chr16:14,935,711-14,954,790) to each haplotype separately with wfmash (v0.7; parameters: -p 80 –num-mappings-for-segment = 10000) and minimap2 (v2.22; parameters: -x map-ont -f 5000 -N 300 -p 0.5),[106,107] restricting to aligned regions of at least 15 kbp. We also applied DupMasker (v1.11)[12] to identify the LCR16a duplicon where *NPIP* is located (SD9443). DupMasker identified additional copies of *NPIP* only in nonhuman primates but was not necessary for detecting *NPIP* copies in human haplotypes. For copies not represented on GRCh38, we used phylogenetic grouping (see below) to further identify paralogs that were unique to some haplotypes but not present in others (i.e., L1, etc.).

## Cell Genomics
### Article

### Other gene annotation

We annotated genes on each haplotype with Liftoff (v1.6.3; parameters: -flank 0.1 -polish -sc 0.85 -copies -mm2_options = "-a –end-bonus 5 –eqx -N 10000 -p 0.3 -f 1000"),[99] using protein-coding genes in GENCODE v44[108] on GRCh38 as the reference annotation set.

### Phylogenetic paralog identity

We created an MSA of *NPIP* genes from each human and *Pongo pygmaeus* haplotype using MAFFT (v7.487; FFT-NS-2).[20] To create a phylogenetic tree of *NPIP* paralogs, we trimmed VNTRs, exons, and poorly aligned regions from the MSA visually. We estimated a maximum-likelihood phylogeny from this MSA using IQ-TREE (v2.2.3 COVID-edition; parameters -B 1000 -alrt 1000) with the GTR+F+R6 substitution model selected with ModelFinder.[109,110] Ultrafast bootstrap and SH-aLRT were used as measures of clade confidence.[101,111] Clades were named based on annotations of GRCh38, T2T-CHM13 v2.0, and T2T-HG002 and defined with SH-aLRT branch support values > 75.

### Assembly validation

Assembled regions were validated with NucFreq, Flagger, and GAVISUNK depending on availability of orthogonal sequencing data.[16,41,47,112] Regions with no read support, only ONT support, or only HiFi support were removed. Flagger and NucFreq were applied to hifiasm and Verkko assemblies and excluded erroneous, falsely duplicated, collapsed, low confidence, or unreliable blocks. GAVISUNK was applied to hifiasm assemblies as described in Vollger, 2023, and supported regions were kept for downstream analyses.[48] Only assemblies that were contiguous between proximal and distal non-segmentally duplicated marker genes were considered.

### Locus configuration comparisons

To compare structural configurations for each *NPIP* locus across samples, 10 loci were defined based on adjacent non-duplicated genes from Liftoff annotations. Configurations were defined based on order and orientation of *NPIP* paralogs and protein-coding genes from the Liftoff annotations relative to adjacent marker genes. Only configurations that passed assembly validation in at least one haplotype and were detected in at least two haplotypes were considered for further analysis. To calculate rearrangement distance between each configuration at each locus, the order and orientation of DupMasker annotations of at least 1 kbp and protein-coding marker genes were used as input to the capping-free double-cut-and-join indel model,[71] and the matrix of pairwise rearrangement distances was transformed into a midpoint-rooted neighbor-joining tree with Bio.Phylo.[113] The resulting trees, gene annotations, and DupMasker content were visualized with custom scripts and Baltic.[114]

### VNTR analysis

To measure the length of *NPIP* exon 8 VNTRs, Tandem Repeats Finder (TRF v4.10; parameters 2 5 7 80 10 10 2000 -d -ngs) was applied to each *NPIP* copy from each haplotype.[115] The longest contiguous region of tandem repeats with period of at least 40 bp was considered for each *NPIP* copy. Exon 8 VNTR size was also called directly from Iso-Seq predicted ORFs by counting substrings containing "SADD" and "ISR" for the two frames of the repeat.

### Gene model and ORF prediction

Iso-Seq reads were used to generate gene models on each human haplotype with PacBio Pigeon and SQANTI3 (v5.2), and ORF sequences with GeneMark.[116,117] Only uniquely-mapping Iso-Seq reads were used for gene model prediction, defined as a delta of at least one additional mismatch between the best-mapping paralog and second-best mapping. Mono-exonic reads were excluded. For comparison to gene models, ORFs were called directly from each *NPIP* Iso-Seq read with ANGEL,[118] keeping the longest ORF per molecule.

### Selection analyses

PAV v2.4.0.1 was used to call variants for each assembled HGSVC3 (Freeze 4) haplotype relative to T2T-CHM13 v2.0.[15] Analysis was restricted to chromosome 16, containing all but one *NPIP* paralog, and African samples ($n = 20$) to reduce the impact of population bottlenecks in the human demography. Variants were restricted to biallelic SNPs with BCFtools.[119] Tajima's D was estimated for sliding 30 kbp windows with VCF-kit ($n = 510,732$ windows for chromosome 16 (96 Mbp).[120] For comparison, Tajima's D was called in the same way using high-coverage Illumina data for Gambian samples in the 1KG samples ($n = 119$), restricting to 95% mappable regions as defined by the Genome in a Bottle Consortium.[121] $nS_L$ was called for 30 kbp windows with selscan v2.0.2,[74] for PAV (long-read) and 1KG (short-read) samples. PAV African $nS_L$ results were jointly normalized for variant frequency with 95% mappable short-read calls for Gambian (GWD) samples (parameters –nsl –bins 100 –qbins 10 –min-snps 10 –bp-win –winsize 30000). Windows overlapping a T2T-CHM13 v2.0 *NPIP* copy by at least 5 kbp were considered valid.

### Protein structure prediction

For the long exon 8 VNTR isoforms predicted with SQANTI3, protein structures were predicted with Chai-1, using MSA-free mode as *NPIP* does not have the deep homology exploited by MSA-based methods for structure prediction.[102] Protein structure predictions were visualized with ChimeraX.[122]

### Short-read RNA-seq expression analysis

To quantify *NPIP* paralog-specific expression from short-read RNA-seq, reads were first aligned to T2T-CHM13 v2.0 with hisat2.[103] Jellyfish was used to find all possible 31-mers from *NPIP* gene models that were not found in the rest of the T2T-CHM13 v2.0 genome.[100] The uniqueness of each k-mer was classified by the number of T2T-CHM13 *NPIP* paralogs in which it was found, and paralogs were iteratively merged to form detectable paralog groups until each group contained at least five uniquely identifying k-mer positions. A custom script was then used to count each identifying k-mer with each RNA-seq read and classify reads by paralog group.

### Visualization

Structural variant and phylogenetic visualizations were created with SVbyEye, archaeopteryx, augur, MEGA, and augur/auspice.[123–126]

### Timetree analysis

A timetree was inferred by applying the LSD2 method[43] to a maximum likelihood neutral *NPIP* phylogenetic tree estimated with IQ-TREE,[109] including a single human sequence for each paralog extracted from CHM13, CHM1, GRCh38, HG002, or PNG15, along with ape sequences from the primary *Pan troglodytes*, *Pan paniscus*, *Gorilla gorilla*, *Pongo pygmaeus*, *Pongo abelii*, and *Symphalangus syndactylus* haplotypes from the T2T Primate Project (v2.0),[44] and the ancestral *NPIP* gene from *Macaca fascicularis*[45] as the outgroup, aligned with MAFFT.[20] The timetree was calibrated with the *Homo-Macaca* divergence set as 28.8 mya. The estimated log likelihood value of the tree is −91,940.323. There were 11,202 total positions in the final dataset.

### Probe design, cDNA generation, enrichment, and sequencing

Biotinylated oligonucleotide probes targeting *NPIP* (Table S3) were designed to enrich in *NPIP* FLNC cDNA as described in Dougherty, 2018. Briefly, probes were designed to target constitutive exons for subfamilies A and B (exons 2, 3, 5, 6, and 7), avoiding repeat-masked sequence. Then, 5′ biotinylated sense strand oligonucleotides were synthesized by IDT for *NPIP* enrichment.

cDNA were generated using the Clontech SMARTer PCR cDNA Synthesis Kit for CHM1 (BioSample SAMN02205338), adult brain (Clontech catalog no. 636102), fetal brain (Clontech catalog no. 636106), heart (Takara catalog no. 636532, lot 1902102A), lung (Origene sample ID FR5B3386C1), ovary (Origene sample ID FR00027E9B), thymus (Origene sample ID FR5B338054), and testis (Takara catalog no. 636533 lot 1402004; BioSample SAMN15935045). For fetal brain and testis samples, cDNA were also generated using the TeloPrime Full-Length cDNA Amplification Kit V2 (Lexogen), which aims to avoid generating truncating cDNA by requiring the 5′ mRNA cap.

Unenriched polyA cDNA were sequenced from heart, lung, ovary, and thymus samples, which were barcoded and pooled on a PacBio Sequel II SMRT cell with 30-h movie time and two-hour pre-extension.

Hybridization capture was performed on cDNA from the remaining tissues using the biotinylated *NPIP* probes, as described in Dougherty, 2018. A single Sequel SMRT cell was used for each of CHM1 and adult brain, with the remaining samples barcoded and pooled for sequencing.

Iso-Seq data from previous publications and public data depositions were obtained from ANVIL, ENCODE, and SRA, as referenced in Table S1, and analyzed together with our generated FLNC data.[22–36]

## QUANTIFICATION AND STATISTICAL ANALYSIS

Statistical analyses were performed as described in the legends for Figures 1 and 6, with SciPy. We assessed significance with a nominal *p*-value cutoff of 0.05.

