## [Document S2. Transparent peer review records for Dishuck et al · Cell Genomics]

Structural variation, selection, and diversification of the NPIP gene family from the human pangenome

Author list: Philip C. Dishuck, Katherine M. Munson, Alexandra P. Lewis, Max L. Dougherty, Jason G. Underwood, William T. Harvey, PingHsun Hsieh, Tomi Pastinen, Evan E. Eichler¹,

Summary

Initial submission: Received : February 27th 2025

Scientific editor: Judith Nicholson

First round of review: Number of reviewers: 3
Revision invited : April 4th 2025
Revision received : June 9th 2025

Second round of review: Number of reviewers: 3
Accepted : 24th July 2025

Data freely available: Yes

Code freely available: N/A

This transparent peer review record is not systematically proofread, type-set, or edited. Special characters, formatting, and equations may fail to render properly. Standard procedural text within the editor's letters has been deleted for the sake of brevity, but all official correspondence specific to the manuscript has been preserved.

Referees' reports, first round of review

Reviewer 1:

This is the most thorough characterization of the NPIP gene family that has so far been generated. The study provides detailed insights into several important characteristics of the NPIP family, including its genome organization, evolution, expression and variation, and also highlights the strong evolutionary selection pressures that act on it. As such, it will be a highly valuable resource for future research into this complex and dynamic gene family.

There are a number of other important highly duplicated sequences in the human and non-human primate genomes, both variant and invariant, and this state-of-the-art study will be a model for future investigations that attempt to effectively characterize such challenging parts of the genome.

Reviewer 2:

The paper by Dishuck, et al. describes detailed analysis of the expansion of the NPIP gene family based on extensive evaluation of long-read sequencing data from multiple cohorts and datasets at both the genomic and transcriptome level.

Overall, the manuscript is clearly written and conveys key features of the complex structure and inferred history of these loci.

The authors note biases in IGC (top of p. 5) that may be due to functional constraint and/or bias in the mutational process. Are there other loci that these patterns can be compared to? Is this observation generalizable?

Considerable attention is paid to analysis of full-length NPIP copies, their transcripts and ORF structures. It seems that all named/numbered NPIP copies are believed to be active genes (please confirm). A few that had previously been considered pseudogenes are mentioned, but it would be good to add an analysis/discussion the extent and nature of pseudogenes based on the improved assembly and annotation data presented here. If there are no pseudogene copies, what does that say about functional constraint of the active copies?

Many other genes are components of the SD regions described here (Figure 3). The authors should address copy number variation of these genes across haplotypes as well. It is noted that the LCR16 duplications have been associated with neurodevelopmental delay in many studies. The extent to which this is fully attributable to NPIP gene variation is probably still an open question, but with the investment described here in creating fully phased haplotypes across the region, it would be good to see the extent of genic variation that is present outside the NPIP family as well.

p. 8 - "expanded among the African great apes" -- have there been independent expansions in these lineages? Certainly the available non-human data is much less complete, but any insights would be helpful in understanding whether expansion is likely to be unique to the human lineage.

Minor points:

- o line numbers in the MS would have been helpful

- o p. 4 - why were reads mapped to the NPIP locus from GRCh38, which seems like the least useful of available references?

- o p. 5 in the discussion of 13 structural configurations - can you tell whether inversions are linked to IGC or are independent of it?

p. 8 - NPIP9 and B15 are noted to be among the top 5% for positive selection in at least one test, but that is limited to consideration only of chr16 genes. p. 6 suggests they are in the top 1% genome-wide, which is quite a different way of describing this. Please be consistent.

Links to data in the first three rows of Supplemental Table 1 do not seem to work.

Figure 1B - what does the asterisk on the African label denote?

Figure 4B/C/D -- is the secondary axis with range 0-1 really "Percent of SNPs..." or should it be "Fraction of...."

Reviewer 3:

The authors have leveraged high-quality human genome assemblies from the pangenome project and the HGSC (total of 169 haplotypes) to characterize genetic diversity in the NPIP gene family. This builds on previous work published long time ago in 2001 that first identified the NPIP gene family and demonstrated evidence for positive selection. The authors have performed comprehensive analysis of NPIP genetic diversity and integrated the genomic data with long-read Iso-Seq transcriptomic data to understand the expression patterns and transcript variation of the different paralogs of the NPIP gene family. The paper is well written and the figures are quite comprehensive. Overall, the paper significantly adds to our understanding of this gene family. Nevertheless, the function of the gene family and its different members still remains elusive. My main comments about the paper are as follows:

1. The authors use Tajima's D and another selection test for extended haplotype homozygosity to find that many different paralogs of this family show evidence of both positive and balancing selection. This is interesting and consistent with previous work showing that the NPIP family members show evidence of positive selection in the coding regions. However, the statistical analysis was performed using only 20 African ancestry samples which is likely to limit the power to detect selection. Also, as the authors pointed out, the NPIP family shows extensive gene conversion which is likely to affect selection tests such as Tajima's D and mimic the effects of natural selection. As the authors showed that the D statistic distribution is changed when IGC-overlapping windows are included. Overall, the evidence for claiming both positive and balancing selection can be bolstered by showing that well-known loci under selection are detected from the 20 samples (power) and (ii) Tajima's D can detect selection even in the presence of IGC.

2. Another concern is regarding the assembly validation rate of the different NPIP paralogs which varies from 52-85% (Figure 2). It is understandable that even with long read sequencing, there are still some errors in the assemblies. The question is whether these errors can impact the downstream analysis

(genetic diversity, selection tests, etc) and to what extent. Some comments on this would be helpful.

Minor comments:

The number of windows used for selection analysis should be indicated.

The DNA sequences of the different NPIP genes should be provided with the paper or made available in a public repository.

Authors' response to the first round of review

Blue: Reviewer Text

Black: Response

Reviewer #1: This is the most thorough characterization of the NPIP gene family that has so far been generated. The study provides detailed insights into several important characteristics of the NPIP family, including its genome organization, evolution, expression and variation, and also highlights the strong evolutionary selection pressures that act on it. As such, it will be a highly valuable resource for future research into this complex and dynamic gene family.

There are a number of other important highly duplicated sequences in the human and non-human primate genomes, both variant and invariant, and this state-of-the-art study will be a model for future investigations that attempt to effectively characterize such challenging parts of the genome.

We thank the reviewer for their encouraging comments and recognition of the broader utility of the approach for solving other gene-rich dynamic regions of the genome.

Reviewer #2: The paper by Dishuck, et al. describes detailed analysis of the expansion of the *NPIP* gene family based on extensive evaluation of long-read sequencing data from multiple cohorts and datasets at both the genomic and transcriptome level.

Overall, the manuscript is clearly written and conveys key features of the complex structure and inferred history of these loci.

We thank the reviewer for their encouraging comments and feedback.

The authors note biases in IGC (top of p. 5) that may be due to functional constraint and/or bias in the mutational process. Are there other loci that these patterns can be compared to? Is this observation generalizable?

While we are still working on a genome-wide analysis, we have begun to observe some striking examples of biased directionality of IGC in other recent functional duplicate loci. For example, *NOTCH2NLA* and *NOTCH2NLB* are both recent duplicates implicated in delayed neuronal maturation and adaptive evolution of the human brain (Fiddes et al., 2019). Analyzing completely sequenced human haplotypes, we found a preferential directional effect for gene conversion where *NOTCH2NLA* converts *NOTCH2NLB* but never in the reverse direction (Real et al., bioRxiv, 2025). Thus, we believe it is more generalizable and it may be one potential approach to pinpoint duplicate genes under constraint or functionally more important paralogs. We added two sentences to this effect in the discussion:

"Based on the human genomes we surveyed here, only the *NPIP2* locus is invariant by all analyses and counter to expectation, *NPIP2* is the only paralog where all predicted ORFs are truncated and may be a *bona fide* pseudogene. Notably, this particular copy is the only paralog that is isolated (i.e., not found in a cluster with other *NPIP* paralogs nor associated with other flanking SDs). This stands in contrast to the ancestral locus, *NPIPA1*¹⁻³, which is both copy number polymorphic and shows a striking asymmetry for IGC, serving only as a donor and never an acceptor of a gene conversion event. **We similarly observed asymmetric IGC in another recent human duplication, the *NOTCH2NL* gene family, which is associated with expansion of the human neocortex. In that case, 20% of haplotypes converted *NOTCH2NLB* to *NOTCH2NLA*, while the reciprocal event is never observed**⁸⁴. Such striking patterns of biased IGC may further pinpoint the copies most likely to confer function."

Considerable attention is paid to analysis of full-length *NPIP* copies, their transcripts and ORF structures. It seems that all named/numbered *NPIP* copies are believed to be active genes (please confirm). A few that had previously been considered pseudogenes are mentioned, but it would be good to add an analysis/discussion the extent and nature of pseudogenes based on the improved assembly and annotation data presented here. If there are no pseudogene copies, what does that say about functional constraint of the active copies?

In our analysis, *NPIP2* does not maintain a full-length ORF after consideration of long-read transcriptomic data (FLNC). It is the only copy where the ORF appears truncated in all haplotypes. We speculate that widespread IGC among *NPIP* paralogs may expose the remaining copies to preferential or directional homogenization allowing disruptive CDS mutations to quickly spread. A disruption of their ORFs has the potential to endanger the activity or function of other paralogs to create dominant negative interactions with other proteins as originally postulated (Johnson et al., 2001). Thus, constrained paralogs readily serve as a template to repair damaged paralogs through IGC. It is noteworthy that *NPIP2* is

relatively isolated. It is one of only three *NPIP* paralogs that is an "orphan" located far away from other *NPIPs* instead of within a cluster. It is the only paralog not associated with other flanking SDs—both properties make it less subject to potential IGC.

We revised the second paragraph of the discussion to reflect this as follows:

"Because functional human-specific duplicate genes have been shown to be frequently invariant^{62,63,83}, we systematically assessed copy number for each paralog. Based on our 28 distinct human phylogenetic groups, we find only three loci that are copy number invariant (*NPIP2*, *B11*, and *B14*). We also distinguish loci that always have at least one copy in humans although often more (*A2*, *A4*, *B12/B13*, and *B15*). It should be noted that several of these copies are associated with larger scale structural changes, such as inversion polymorphism or IGC events. Based on the human genomes we surveyed here, only the *NPIP2* locus is invariant by all analyses and **counter to expectation, *NPIP2* is the only paralog where all predicted ORFs are truncated and may be a bona fide pseudogene. Notably, this particular copy is the only paralog that is isolated (i.e., not found in a cluster with other *NPIP* paralogs nor associated with other flanking SDs).** This stands in contrast to the ancestral locus, *NPIPA1*¹⁻³, which is both copy number polymorphic and shows a striking asymmetry for IGC, serving only as a donor and never an acceptor of a gene conversion event. We similarly observed asymmetric IGC in another recent human duplication, the *NOTCH2NL* gene family, which is associated with expansion of the human neocortex. In that case, 20% of haplotypes converted *NOTCH2NLB* to *NOTCH2NLA*, while the reciprocal event is never observed⁸⁴. Such striking patterns of biased IGC may further pinpoint the copies most likely to confer function."

Many other genes are components of the SD regions described here (Figure 3). The authors should address copy number variation of these genes across haplotypes as well. It is noted that the LCR16 duplications have been associated with neurodevelopmental delay in many studies. The extent to which this is fully attributable to *NPIP* gene variation is probably still an open question, but with the investment described here in creating fully phased haplotypes across the region, it would be good to see the extent of genic variation that is present outside the *NPIP* family as well.

This is an interesting suggestion and we performed an analysis where we assessed all SDs and unique sequence flanking *NPIP* copies in the disease regions defined in Figure 3 (chr16p13.12-p11.2), using gene definitions to define the unit for comparison among the haplotypes. We examined 102 haplotypes (highly contiguous HiFi + ONT assemblies from HGSC year 3) and 105 flanking SD-associated genes and found that 63% show evidence of copy number variation. If we restrict this analysis to genes mapped to SD regions in T2T-CHM13, this increases to 81%. For example, *SULT1A1*—a cytosolic phenol sulfotransferase enzyme—is predicted to vary in copy number from 1-3 copies per haplotype (see below and Figure 3G).

We added a note describing this new analysis to the results as follows:

"Notably, we find that many of the human haplotypic configurations occurred in conjunction with copy number variation and IGC events associated with specific *NPIP* loci. **A complete assessment of copy number variation of both "unique" and SD genes flanking *NPIP* in disease regions may be found in the supplement (Table S4 and Figure S7).**"

Figure S7. Estimated CN of genes in *NPIP* loci from 102 haplotypes. Genes within 30 kbp of an *NPIP* SD block (chr16p13.12-p11.2) are included. Only variable genes shown and 66/105 genes show copy number variation. Rows correspond to genes in chromosome order, and each column represents an assembled sample, with the two haplotypes summed. See Figure 1 for locations of some of the SD genes and Table S4 for detailed breakdown.

p. 8 - "expanded among the African great apes" – have there been independent expansions in these lineages? Certainly the available non-human data is much less complete, but any insights would be helpful in understanding whether expansion is likely to be unique to the human lineage.

Yes, based on phylogenetic reconstruction and comparative sequence analysis, it has been shown that the *NPIP* gene family has expanded independently at least six times over the last 50 million years of primate genome evolution. Though not a focus of this manuscript, independent expansions have been documented in chimpanzee, bonobo, orangutan, gorilla, and marmoset (a New World monkey) in addition to human (Johnson 2001 and 2006, Cantsilieris 2020, Damert 2022, Yoo 2025). In some cases, duplications have been transposed to chromosomes other than those syntenic to chromosome 16. For example (Figure S1), orangutans have additional duplications on chromosomes 11, 12, and 13, chimpanzee and bonobo on chromosomes 7 and 17, and gorilla on chromosome 19, not seen in human samples. In all cases, the spread of *NPIP* has been associated with additional SDs often specific to the lineage in question. Thus, the flanking SDs become diagnostic for different evolutionary trajectories of *NPIP* (see Cantsilieris, 2020).

Figure S1. Timetree of human and ape *NPIP* duplications. The estimated age of *NPIP* paralogs for humans and ape species *Pan troglodytes* (Ptr), *Pan paniscus* (Ppa), *Gorilla gorilla* (Ggo), *Pongo pygmaeus* (Ppy), *Pongo abelli* (Pab), *Symphalangus syndactylus* (Ssy) is shown on a neutral phylogeny. The tree is rooted to the single copy ancestral *NPIP* from *Macaca fascicularis* (Mfa), with divergence time set to 28.8 mya. Human paralogs are bolded, and nonhuman primate paralogs are labeled by species abbreviation, chromosome number (hsa: human homologous chromosome, chr: species chromosome name), and position within the chromosome. Branch time estimates are indicated at the branch point.

Minor points:

o line numbers in the MS would have been helpful

Line numbers have been added to the revised manuscript – we apologize for this oversight.

o p. 4 - why were reads mapped to the NP1P locus from GRCh38, which seems like the least useful of available references?

To be clear, GRCh38 was used primarily for the purpose of gene annotation. We did not map *NP1P* reads to GRCh38; instead, we aligned extracted *NP1P* copies from each assembly to identify the best map location from GRCh38 in order to name the individual gene copies. We preferred to use the gene names from GRCh38 annotations to avoid confusion and to have consistency with prior published work. Whenever new genes were encountered, we examined T2T-CHM13/HG002 assemblies and then subsequently named additional copies based on phylogenetic assignment with the designations (L1, etc.). Moreover, our study began before the completion and annotation of the T2T-CHM13 assembly; however, the ancestral *NP1PA1* was contiguously assembled in GRCh38 from BAC clone RP11-680G24.

Based on the referee's comment, we realize that our description in the Methods was lacking these details and we revised as follows:

***NP1P* gene identification.** To identify *NP1P* gene locations within assemblies, we aligned the ancestral *NP1P* locus from GRCh38 (chr16:14,935,711-14,954,790) to each haplotype separately with wfmash (v0.7; parameters: -p 80 --num-mappings-for-segment=10000) and minimap2 (v2.22; parameters: -x map-ont -f 5000 -N 300 -p 0.5) (Marco-Sola et al. 2023; Li 2018), restricting to aligned regions of at least 15 kbp. We also applied DupMasker (v1.11) (Jiang et al. 2008) to identify the LCR16a duplication where *NP1P* is located (SD9443). DupMasker identified additional copies of *NP1P* only in nonhuman primates but was not necessary for detecting *NP1P* copies in human haplotypes. **For copies not represented on GRCh38, we used phylogenetic grouping (see below) to further identify paralogs that were unique to some haplotypes but not present in others (i.e. L1, etc.).**

Other gene annotation. We annotated other non-*NP1P* genes on each haplotype with LiftOff (v1.6.3; parameters: -flank 0.1 -polish -sc 0.85 -copies -mm2_options="-a --end-bonus 5 --eqx -N 10000 -p 0.3 -f 1000") (Shumate and Salzberg 2021), using protein-coding genes in GENCODE v44 (Frankish et al. 2023) on GRCh38 as the reference annotation set.

Phylogenetic paralog identity. We created an MSA of *NP1P* genes from each human and *Pongo pygmaeus* haplotype using MAFFT (v7.487; FFT-NS-2) (Katoh and Standley 2013). To create a phylogenetic tree of *NP1P* paralogs, we trimmed VNTRs, exons, and poorly aligned regions from the MSA visually. We estimated a maximum-likelihood phylogeny from this MSA using IQ-TREE (v2.2.3 COVID-edition; parameters -B 1000 -alrt 1000) with the GTR+F+R6 substitution model selected with ModelFinder (Minh et al. 2020; Kalyaanamoorthy et al. 2017). Ultrafast bootstrap and SH-aLRT were used as measures of clade confidence (Hoang et al. 2018; Anisimova et al. 2011). Clades were named based on annotations of GRCh38, T2T-CHM13 v2.0, and T2T-HG002 and defined with SH-aLRT branch support values >75.

o p. 5 in the discussion of 13 structural configurations - can you tell whether inversions are linked to IGC or are independent of it?

We assessed inversion status and IGC independently. Based on the referee's suggestion, we analyzed 65 inverted haplotypes at 16p11.2 (*NP1PB6*, *B7*, *B8*, *B9*) and determined that of these 6 (9%) showed evidence of IGC. By contrast, all 40 direct orientation haplotypes (including the T2T-CHM13 reference's

D7 haplotype) showed evidence of gene conversion relative to the most common haplotype, I9 (corresponding to GRCh38), in that *NPIP* paralogs were situated in non-syntenic positions that are not possible outcomes of inversions. For example, in direct orientation haplotypes *B7* is always present at one or both edges of the locus despite most commonly being situated internally. Nuttle et al. created a model for the structural evolution of 16p11.2 in great apes, predicting that the inverted orientation is ancestral (Nuttle et al., 2016) but has toggled orientation at least twice as part of complex rearrangements. Our findings indicate that *NPIP* gene diversification likely occurred in the inverted state for this locus.

We added text to the results summarizing this finding as follows:

"The chromosome 16p11.2 locus contains *NPIP*B6, *B7*, *B8*, and *B9* spanning a 650 kbp inversion polymorphism (Fig. 3D,G). Through IGC and inversions, the *B7* sequence can occupy any of the four canonical *NPIP* locations in this cluster—thus effectively "relocating" or "repositioning" as a result of IGC. Additionally, 8/13 configurations also have a 355 kbp inversion with respect to T2T-CHM13, and only the inverted orientation configurations carry the *B6* or *B9* genes. At 99.6% sequence identity to the reference genome, the I9 inverted region is among the most divergent (top 9.5%) euchromatic regions of the human genome. **Notably, all direct orientation haplotypes (relative to T2T-CHM13) exhibit *NPIP* repositioning best explained by IGC, in contrast to 9% of inverted haplotypes, perhaps indicating that these paralogs underwent a period of diversification in the inverted orientation.** We also observed 1.6 and 1.3 Mbp inversions at chromosome 16p12.3 and p12.2, respectively (Fig. 3D,E). Altogether, of the nine loci containing *NPIP* paralogs, only the locus at chromosome 16p13.3 containing *B2* is structurally invariant."

p. 8 - *NPIP*B9 and *B15* are noted to be among the top 5% for positive selection in at least one test, but that is limited to consideration only of chr16 genes. p. 6 suggests they are in the top 1% genome-wide, which is quite a different way of describing this. Please be consistent.

We apologize for the confusion, but we only estimated selection values for chromosome 16 as described in the legends and methods. However, we now make this clearer in the results as well:

"To assess whether positive selection is still ongoing in the human population and narrow down signatures to individual loci, we performed complementary tests of Tajima's *D* and nS_L for extended haplotype homozygosity (Tajima 1989; Ferrer-Admetlla et al. 2014) **restricting our analysis to chromosome 16.**"

Links to data in the first three rows of Supplemental Table 1 do not seem to work.

The Zenodo repository is temporary, pending final accession of these data at standard genomics databases. Please note two of the samples are not 1KG open access and, thus, the Papuan Melanesian are required to be placed under restricted access in EGA. Until then, we made the results available to reviewers at:

https://zenodo.org/records/14941578?preview=1&token=eyJhbGciOiJIUzUxMiJ9.eyJpZCI6IjA0ZmEyZGJlLTE1MTItNDNjMj04OWNiLTUxMDg2MmRlZmMzMzMyImlmRhdGEiOnt9LCJyZW5kb20iOiIyOWZkYmU4MGJlNTJkZDljZTk3Mzc3MTYyYiYzNTg3NiJ9.oBJ0zAZbaaNLq_wLxAyq79cBL-jQovM4waEwQRw27kvnZqiam-X4nr-T_LGz70qqSVovK5KmatO1jfQlc5jxvQ. We requested that the link to this private repository be shared with reviewers—our apologies if this was unclear.

Figure 1B - what does the asterisk on the African label denote?

This asterisk indicates a significant increase in copy number among Africans relative to non-African samples. We added additional text to the legend to clarify:

"Copy number is significantly higher in African compared to non-African samples (Wilcoxon rank-sum test, $p=0.000001$)."

Figure 4B/C/D -- is the secondary axis with range 0-1 really "Percent of SNPs..." or should it be "Fraction of...."

Thank you for catching this inaccurate axis title. We adjusted the figure to read "Fraction of SNPs significant for nSL".

Reviewer #3: The authors have leveraged high-quality human genome assemblies from the pangenome project and the HGSCV (total of 169 haplotypes) to characterize genetic diversity in the NPIP gene family. This builds on previous work published long time ago in 2001 that first identified the NPIP gene family and demonstrated evidence for positive selection. The authors have performed comprehensive analysis of NPIP genetic diversity and integrated the genomic data with long-read Iso-Seq transcriptomic data to understand the expression patterns and transcript variation of the different paralogs of the NPIP gene family. The paper is well written and the figures are quite comprehensive. Overall, the paper significantly adds to our understanding of this gene family. Nevertheless, the function of the gene family and its different members still remains elusive. My main comments about the paper are as follows:

Thank you for taking the time to review our paper and provide feedback.

1. The authors use Tajima's D and another selection test for extended haplotype homozygosity to find that many different paralogs of this family show evidence of both positive and balancing selection. This is interesting and consistent with previous work showing that the NPIP family members show evidence of positive selection in the coding regions. However, the statistical analysis was performed using only 20 African ancestry samples which is likely to limit the power to detect selection. Also, as the authors pointed out, the NPIP family shows extensive gene conversion which is likely to affect selection tests such as Tajima's D and mimic the effects of natural selection. As the authors showed that the D statistic distribution is changed when IGC-overlapping windows are included. Overall, the evidence for claiming both positive and balancing selection can be bolstered by showing that well-known loci under selection are detected from the 20 samples (power) and (ii) Tajima's D can detect selection even in the presence of IGC.

As a preponderance of prior human research has used European samples to discover signature of selection, we used the recently released European samples from the Human Pangenome Reference Consortium's Release 2 (https://github.com/human-pangenomics/hprc_intermediate_assembly/tree/main/data_tables) in combination with HGSCV3 and the HG002-T2T project to assess known loci that are well established for positive selection. We specifically limited our analysis to 20 European samples (40 haplotypes) to assess power of detection.

Prior simulations estimate Tajima's D to have 60% power for detecting selection coefficients of 0.0001 or stronger with 20 samples (Simon and Huttley 2021; doi:10.1101/2021.07.04.451068). We tested Tajima's D on the 40 European haplotypes and compared positive selection signatures identified from 97 European samples where short-read sequence data were available as our positive control (Pybus et al. 2013 NAR; doi:10.1093/nar/gkt1188). The larger sample SRS (short-read sequencing) data detected positive selection at three pigmentation-related loci: *SLC45A2*, *HERC2*, and *SLC24A5*, but surprisingly not at the well-known *LCT* locus. With 20 European phased genome samples from long-read assemblies, we replicated positive selection signatures at *HERC2* and *SLC24A5*, but not at *SLC45A2*, their weakest signal based on average Tajima's D ($D = -1.60$) (see table below).

Gene	Short-read European samples (n=97) (Pybus et al., 2013)		Long-read assemblies from European (n=20)	
	Length Selected Region (kb)	Average Tajima's D	Length Selected Region (kb)	Average Tajima's D
SLC45A2	57	-1.60	-	-0.37
HERC2	225	-1.97	279	-2.11
SLC24A5	144	-2.29	108	-2.81

2. Another concern is regarding the assembly validation rate of the different NPIP paralogs which varies from 52-85% (Figure 2). It is understandable that even with long read sequencing, there are still some errors in the assemblies. The question is whether these errors can impact the downstream analysis (genetic diversity, selection tests, etc) and to what extent. Some comments on this would be helpful.

Like the reviewer, we were concerned that misassembly and IGC could bias Tajima's D. With the same samples (n=20 Europeans), we examined genome-wide Tajima's D distributions as we applied progressive filters, including IGC calls using the methods from Vollger et al., 2023. Misassembled regions and IGC both do distort somewhat the distribution of Tajima's D as shown below; however, most of the effect is driven by misassembly of acrocentric chromosomes, which is not an issue for chromosome 16. Chromosome 16 shows negligible effect for misassembly but is more relevant for IGC regions. Moreover, eight of 11 paralogs retained significance for tests of positive selection (most extreme 5% of windows) when we considered selection signatures outside of detected IGC tracts (see Figure S8). For this reason, we excluded misassembled regions (Figure 4) as well as IGC in the supplement (Figure S8).

Figure S8. Effect of assembly filtering on Tajima's D distribution. A-C) Tajima's D distributions per chromosome from 20 long-read European assemblies (40 haplotypes). A) No filtering applied. B) Uncallable (ambiguously aligned) regions and misassemblies removed. C) IGC tracts removed. D) Genome-wide Tajima's D with (black) and without (red) IGC tracts included and limited to only regions of IGC (green).

Minor comments:

The number of windows used for selection analysis should be indicated.

We added this detail to the methods. Tajima's D was estimated for sliding 30 kbp windows with VCF-kit (Cook and Andersen 2017) (**n= 510,732 windows for chromosome 16 (96 Mbp)**).

The DNA sequences of the different NPIP genes should be provided with the paper or made available in a public repository.

The cDNA and predicted protein sequences for *NPIP* paralogs/isoforms are now available in Table S5 and will be submitted to GenBank with the final paper.

Referees' report, second round of review

Reviewer 1:

Accept

Reviewer 2:

Accept

Reviewer 3:

The author have addressed the comments from the previous review satisfactorily.
I have no further comments.

Authors' response to the second round of review